# Online Feature Updates Improve Online (Generalized) Label Shift Adaptation

**Ruihan Wu**[*][†]
UC San Diego
ruw076@ucsd.edu

**Siddhartha Datta**[*][†]
University of Oxford
siddhartha.datta@cs.ox.ac.uk

**Yi Su**
Google DeepMind
yisumtv@google.com

**Dheeraj Baby**
UC Santa Barbara
dheeraj@ucsb.edu

**Yu-Xiang Wang**
UC San Diego
yuxiangw@ucsd.edu

**Kilian Q. Weinberger**
Cornell University
kilian@cornell.edu

## Abstract

This paper addresses the prevalent issue of label shift in an online setting with missing labels, where data distributions change over time and obtaining timely labels is challenging. While existing methods primarily focus on adjusting or updating the final layer of a pre-trained classifier, we explore the untapped potential of enhancing feature representations using unlabeled data at test-time. Our novel method, Online Label Shift adaptation with Online Feature Updates (OLS-OFU), leverages self-supervised learning to refine the feature extraction process, thereby improving the prediction model. By carefully designing the algorithm, theoretically OLS-OFU maintains the similar online regret convergence to the results in the literature while taking the improved features into account. Empirically, it achieves substantial improvements over existing methods, which is as significant as the gains existing methods have over the baseline (i.e., without distribution shift adaptations).

## 1 Introduction

The effectiveness of most supervised learning models relies on a key assumption that the training data and test data share the same distribution. However, this assumption rarely holds in real-world scenarios, leading to the phenomenon of *distribution shift* [41, 2]. Previous research has primarily focused on understanding distribution shifts in offline or batch settings, where a single shift occurs between the training and test distributions [33, 45, 51, 52, 35]. In contrast, real-world applications often involve test data arriving in an *online* fashion, and the distribution shift can continuously evolve over time. Additionally, there is another challenging issue of *missing and delayed* feedback labels, stemming from the online setup, where gathering labels for the streaming data in a timely manner becomes a challenging task.

To tackle the distribution shift problem, prior work often relies on additional assumptions regarding the nature of the shift, such as label shift or covariate shift [43]. In this paper, we focus on the common *(generalized) label shift* problem in an online setting with missing labels [49] . Specifically, the learner is given a fixed set of labeled training data $D_0 \sim \mathcal{P}^{\text{train}}$ in advance and trains a model $f_0$. During test-time, only a small batch of unlabelled test data $S_t \sim \mathcal{P}_t^{\text{test}}$ arrives in an online fashion ($t = 1, 2, \cdots$). For the online label shift, we assume the label distribution $\mathcal{P}_t^{\text{test}}(y)$ may change over time $t$ while the conditional distribution remains the same, i.e. $\mathcal{P}_t^{\text{test}}(x|y) = \mathcal{P}^{\text{train}}(x|y)$. For example, employing MRI image classifiers for concussion detection becomes challenging as label shifts emerge from

---

[*]Equal contribution

[†]Work done while at Cornell University

38th Conference on Neural Information Processing Systems (NeurIPS 2024).

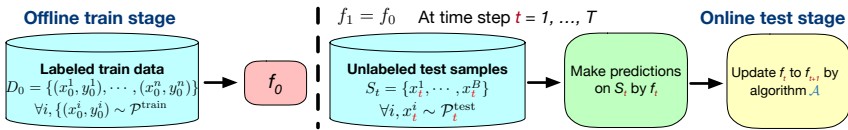

Figure 1: Overview of online distribution shift adaptation. We further assume $\mathcal{P}_t^{\text{test}}(x|y) = \mathcal{P}^{\text{train}}(x|y)$ for online label shift and assume the existence of an unknown feature mapping $h$ such that $\mathcal{P}_t^{\text{test}}(h(x)|y) = \mathcal{P}^{\text{train}}(h(x)|y)$ for online generalized label shift.

seasonal variations in the image distribution. A classifier trained during skiing season may perform poorly when tested later, given the continuous change in image distribution between skiing and non-skiing seasons. In contrast to label shift, the *generalized* label shift relaxes the assumption of an unchanged conditional distribution on $x$ given $y$. Instead, it assumes that there exists a transformation $h$ of the covariate, such that the conditional distribution $\mathcal{P}_t^{\text{test}}(h(x)|y) = \mathcal{P}^{\text{train}}(h(x)|y)$ stays the same. Reiterating our example, consider an MRI image classifier that undergoes training and testing at different clinics, each equipped with MRI machines of varying hardware and software versions. As a result, the images may display disparities in brightness, resolution, and other characteristics. However, a feature extractor $h$ exists, capable of mapping these variations to the same point in the transformed feature space, such that the conditional distribution $\mathcal{P}(h(x)|y)$ remains the same. In both settings, the goal of the learner is to adapt to the (generalized) label shift within the non-stationary environment, continually adjusting the model's predictions in real-time.

Existing algorithms for online label shift adaptation (OLS) primarily adopt one of two approaches: either directly reweighting of the pretrained classifier $f_0$, or re-training only the final linear layer of $f_0$ — while keeping the feature extractor frozen. Recent work [46, 48, 36, 39] have demonstrated the potential for improving feature extractors, even during test-time and in the absence of labeled data. We hypothesize that a similar effect can be harnessed in the context of (generalized) label shift, leading to the idea of *improving feature representation learning during testing*. In online label shift, updating the feature extractor offers two potential advantages. First, it utilizes the additional unlabeled samples, hence enhancing the sample efficiency of the feature extractor. Second, it enables the feature extractor to adapt to label shift, which is crucial to the learning process as the optimal feature extractor may depend on the underlying label distribution. Particularly in generalized label shift scenarios, where the feature transformation $h$ is often unknown, the integration of extra unlabeled test samples facilitates the learning of $h$.

Building upon this insight, this paper introduces the *Online Label Shift adaptation with Online Feature Updates* (OLS-OFU) framework, aimed at enhancing feature representation learning in the context of online label shift adaptation. Specifically, each instantiation of OLS-OFU incorporates a self-supervised learning method associated with a loss function denoted as $l_{\text{ssl}}$ for feature representation learning, and an online label shift adaptation (OLS) algorithm to effectively address distribution shift. By carefully checking the existing OLS methods and SSL methods, we identify three principles for algorithm design: maintain the theoretical guarantee, obey the underlying assumption of the existing OLS methods and fit the required condition of SSL techniques while avoiding heavy additional computational costs. Within the principles, OLS-OFU is designed as three main steps: at each time step, OLS-OFU first executes a revised OLS algorithm and then every $\tau$ steps, OLS-OFU updates the feature extractor through self-supervised learning and subsequently refines the final linear layer.

In addition to its ease of implementation and seamless integration with existing OLS algorithms, OLS-OFU also shows strong theoretical guarantee and empirical performance. Theoretically, we demonstrate that OLS-OFU effectively reduces the loss of the overall algorithm by leveraging self-supervised learning (SSL) techniques to enhance the feature extractor, thereby improving predictions for test samples at each time step $t$. Empirical evaluations on various datasets, considering both online label shift and online generalized label shift scenarios, validate the effectiveness of OLS-OFU. Our OLS-OFU method achieves substantial improvements over existing OLS methods, which is *as significant as the gains existing OLS methods have over the baseline (i.e., without distribution shift adaptations)*. This demonstrates that integrating online feature updates is as effective in solving online distribution shift as the fundamental online label shift method itself. Moreover, the improvement is consistent on various datasets, all existing OLS methods and the various choices of SSL techniques. This consistency underscoring its robustness across different scenarios and its generality to incorporate future OLS methods with more advanced online learning techniques and better SSL techniques.

## 2 Problem Setting & Related Work

We start with some basic notations. Let $\Delta^{K-1}$ be the probability simplex. Let $f : \mathcal{X} \to \Delta^{K-1}$ denote a classifier. Given an input $x$ from domain $\mathcal{X}$, $f(x)$ outputs a probabilistic prediction over $K$ classes. For example, $f$ can be the output from the softmax operation after any neural network. If we reweight a model output from $f$ by a vector $p \in \mathbb{R}^K$, we refer to this model as $g(\cdot; f, p)$ with $g$ denotes the method of reweighting. For any two vectors $p$ and $q$, $p/q$ denotes the element-wise division.

**Online distribution shift adaptation.** The effectiveness of any machine learning model $f$ relies on a common assumption that the train data $D_0$ and test data $D_{\text{test}}$ are sampled from the same distribution, i.e., $\mathcal{P}^{\text{train}} = \mathcal{P}^{\text{test}}$. However, this assumption is often violated in practice, which leads to *distribution shift*. This can be caused by various factors, such as data collection bias and changes in the data generation process. Moreover, once a well-trained model $f_0$ is deployed in the real world, it moves into the testing phase, which is composed of a sequence of periods or time steps. The test distribution at time step $t$, $\mathcal{P}_t^{\text{test}}$, from which test data $x_t$ is sampled, may vary over time. One example is that an MRI image classifier might be trained on MRI images collected during skiing season (which may have a high frequency of head concussions) but tested afterward (when the frequency of concussion is lower). The test stage can last several months until the next classifier is trained. During this test period, the distribution of MRI images may undergo continuous changes, transitioning between non-skiing and skiing seasons.

As the test-time distribution changes over time, the challenge lies in how to adjust the model continuously from $f_{t-1}$ to $f_t$ in an online fashion to adapt to the current distribution $\mathcal{P}_t^{\text{test}}$. We call this problem *online distribution shift adaptation* and illustrate it in Figure 1. Given a total of $T$ steps in the online test stage, we define the average loss for any online algorithm $\mathcal{A}$ through the loss of the sequence of models $f_t$, $t \in [T]$ that are produced from $\mathcal{A}$, i.e.,

$$L(\mathcal{A}; \mathcal{P}_1^{\text{test}}, \cdots, \mathcal{P}_T^{\text{test}}) = \frac{1}{T} \sum_{t=1}^{T} \ell(f_t; \mathcal{P}_t^{\text{test}}), \tag{1}$$

where $\ell(f; \mathcal{P}) = \mathbb{E}_{(x,y) \sim \mathcal{P}} \ell_{\text{sup}}(f(x), y)$ and $\ell_{\text{sup}}$ is the loss function, for example, 0-1 loss or cross-entropy loss for classification tasks.

In this paper, we consider the challenging scenario where at each time step $t$, only *a small batch of unlabeled samples* $S_t = \{x_t^1, \cdots, x_t^B\}$ is received. We formalize the algorithm $\mathcal{A}$ as: $\forall t \in [T]$,

$$f_t := \mathcal{A}\left(\{S_1, \cdots, S_{t-1}\}, \{f_1, \cdots, f_{t-1}\}, D_0, D_0'\right). \tag{2}$$

In contrast to the classical online learning setup, this scenario presents a significant challenge as classical online learning literature usually relies on having access to either full or partial knowledge of the loss at each time step, i.e., $\ell_{\text{sup}}(f(x_t), y_t)$. In this setting, however, only a batch of unlabeled samples is provided at each time step and this lack of access to label information and loss values presents a significant challenge in accurately estimating the true loss defined in Equation 1.

**Online label shift (OLS) adaptation.** Online label shift assumes the marginal distribution of the label $\mathcal{P}_t^{\text{test}}(y)$ changes over time, while the conditional distribution keeps invariant:

$$\forall t \in [T], \mathcal{P}_t^{\text{test}}(x|y) = \mathcal{P}^{\text{train}}(x|y).$$

This assumption is most typical when the label $y$ is the causal variable and the feature $x$ is the observation [43]. The aforementioned example of concussion detection from MRI images fits this scenario, where the presence or absence of a concussion (label) causes the observed MRI image features. Most previous methods tackle this problem through a non-trivial reduction to the classical online learning problem. Consequently, most of the online label shift algorithms [49, 7, 6] study the theoretical guarantee of the algorithm via the convergence of the regret function, which could be either static regret or dynamic regret. In previous studies, the hypothesis class $\mathcal{F}$ of the prediction function $f$ is typically chosen in one of two ways:

1. $\mathcal{F}$ is defined as a family of post-hoc reweightings of $f_0$, with the parameter space comprising reweight vectors. Examples within this category include ROGD [49], FTH [49], and FLHFTL [6].
2. $\mathcal{F}$ is defined as a family of functions that share the same parameters in $f_0$ except the last linear layer, such as UOGD [7] and ATLAS [7].

Notice that the existing OLS methods don't update the feature extractor at the online test stage, but focus on how to leverage advanced online learning techniques to update the remaining part of the model under certain theoretical guarantees. Our method will be orthogonal to them: it will focus on how to leverage the self-supervised learning techniques to update the feature extractor and, therefore, can improve each of them. Our empirical results in fact show that the improvement from the feature extractor update is as significant as the improvement from online learning techniques.

**Online *generalized* label shift adaptation.** In the context of MRI image classification, where head MRI images serve as the feature $x$, variations in software or hardware across different clinics' MRI machines can introduce discrepancies in image characteristics like brightness, contrast, and resolution. In such scenarios, the conditional probability distribution $\mathcal{P}(x|y)$ is no longer invariant. However, when a feature extractor $h$ is robust enough, it can map images into a feature space where the images from different machines have the same distributions $\mathcal{P}(h(x)|y)$ in the transformed feature space. The concept of generalized label shift, as introduced in Tachet des Combes et al. [47], formalizes this by postulating the existence of an unknown function $h$, such that $\mathcal{P}(h(x)|y)$ remains invariant. The primary challenge in this context is to find the underlying transformation $h$. Building upon this, online generalized label shift assumes that there exists an unknown function $h$ such that

$$\forall t \in [T], \mathcal{P}_t^{\text{test}}(h(x)|y) = \mathcal{P}^{\text{train}}(h(x)|y).$$

Thus, to apply the OLS methods for this more general problem, it is important to learn a proper feature extractor as this underlying function $h$.

**Motivations of deploying self-supervised feature updates.** As we reviewed above, OLS methods in the literature don't update the pretrained feature extractor in the online stage. However, the pretrained feature extractor in $f_0$ can be suboptimal for the online test stage in online (generalized) label shift adaptation, because of the three potential reasons:

1. The amount of pretrained data doesn't achieve the learning capacity of the feature extractor structure, i.e. learning with more data can improve the feature extractor.
2. The optimal feature extractors can be different for two different distributions. In our problem, the distribution shifts overtime and the optimal feature extractor can be dynamic too.
3. Particularly in online generalized label shift adaptation, the domain of the data can be dramatically changed between train and test, and the pretrained feature extractor could have unpredictable performance for the test data.

Fortunately, in online (generalized) label shift adaptation problem, the learner receives many unlabeled test samples $S_1 \cup \cdots \cup S_{t-1}$ before the prediction for time step $t$. On the other hand, self-supervised learning is very powerful for representation learning [18, 32, 30, 16, 37, 9, 23, 20, 24] and domain adaptation [46, 48, 36, 39]. In this paper, we will introduce how to deploy these self-supervised learning techniques to the existing OLS methods such that we can still enjoy the theoretical guarantees from online learning techniques and, more importantly, take the advantage of self-supervised learning to achieve better empirical performance.

## 3 Method

In this paper, we introduce a novel online label shift adaptation algorithm OLS-OFU, that leverages self-supervised learning (SSL) to improve representation learning and can be seamlessly integrated with any existing online label shift (OLS) method. By carefully designing how to place the SSL techniques, our algorithm maintains a similar theoretical guarantee from the existing OLS methods, obeys the underlying assumption in existing OLS methods that is important for the effectiveness, and fits the required condition of the SSL techniques while avoiding heavy additional time cost. Through the derived theoretical results, we further understand how the online learning techniques and SSL techniques contribute together to reduce the test loss in the online label shift adaptation problem. Lastly, we demonstrate how the SSL techniques help with the more challenging problem online generalized label shift adaptation.

### 3.1 Three Principles of Algorithm Design.

To combine the step of feature extractor update with any OLS method, the most straightforward thought is to directly insert this step right before or after the step of original OLS in each time step;

see the summarization of the online process in Figure 1. Starting from this thought, there are three remaining questions before finalizing the algorithm: **Q1**: Inserting this step before or after the original OLS step, which option is better? **Q2**: Besides this feature extractor update step, are any other steps also necessary? **Q3**: How frequently (in terms of time step $t$) should we update the feature extractor? We are going to introduce three principles when designing our algorithm, which helps answer the three questions and together lead to our final design.

**Principle 1: maintain the theoretical guarantee.** One main advantage of existing OLS methods is that they have theoretical guarantees for the performance of any unknown label shifts in the online test stage. We would like to keep this advantage after deploying the feature extractor update step. By carefully checking the theoretical analysis of the existing OLS methods, we find that when we deploy the feature extractor update into ROGD, UOGD, or ATLAS, if we would like to update the feature extractor by the unlabeled test samples including $S_t$, it is necessary to execute this update *after* the step of these OLS methods at time $t$ to maintain the similar theoretical guarantee. This is because the main idea of ROGD, UOGD, and ATLAS is to construct an *unbiased* estimator for the gradient $\nabla_f \ell(f_t; \mathcal{P})$ using a batch of samples $S_t \sim \mathcal{P}_t^{\text{test}}$. The estimator has the form of

$$\sum_{y \in \mathcal{Y}} s_t[y] \cdot \nabla_f \mathbb{E}_{(x,y) \sim \mathcal{P}^{\text{train}}} \ell_{\sup}(f_t(x), y), \tag{3}$$

where $s_t$ is an unbiased estimator for the label marginal distribution $q_t$ and it depends on samples $S_t$ as constructed. If the feature extractor update according to $S_t$ happened before the step of ROGD, UOGD or ATLAS, $f_t$ would not be independent of $s_t$ in Equation 3 and this can break their main idea, as illustrated in the following proposition.

**Proposition 1.** *If $f_t$ is not independent of the samples $S_t$, the gradient estimator in Equation 3 is not guaranteed to be an unbiased estimator of the gradient $\nabla_f \ell(f_t; \mathcal{P})$.*

Thus, to maintain a similar theoretical guarantee after updating the feature extractor by the online test samples, *we should insert the feature update involving $S_t$ after the step of original ROGD, UOGD or ATLAS*, and this answers Q1. In the experiment, we will validate the necessity of this design.

**Principle 2: obey the underlying assumption of the existing OLS methods.** We find that the existing OLS methods ROGD, FTH, and FLHFTL opt for the hypothesis $f$ to be a post-hoc reweight of the pre-trained model $f_0$. The underlying assumption behind this design is that $f_0$ is a good approximation of $\mathcal{P}^{\text{train}}(y|x)$. Within this assumption, because $\mathcal{P}^{\text{test}}(y|x)$ is a post-hoc reweight of $\mathcal{P}^{\text{train}}(y|x)$ in the case of label shift [34, 49], reweighting such $f_0$ can approach $\mathcal{P}^{\text{test}}(y|x)$. Therefore, after updating the feature extractor, we expect that the base model, which is to be reweighted, still approximates the training distribution $\mathcal{P}^{\text{train}}(y|x)$. This can be done by *re-training the linear layer under the training data after the feature extractor update*, and this answers Q2.

**Principle 3: fit the required condition of the SSL techniques while avoiding heavy additional computational costs.** Given a set of unlabeled samples $S$, we denote the loss of an SSL technique for a model $f$ as $\ell_{\text{ssl}}(S; f)$ and the update to this model would be in the form of gradient descent: $\theta^{\text{feat}} \rightarrow \theta^{\text{feat}} - \eta \cdot \nabla_{\theta^{\text{feat}}} \ell_{\text{ssl}}(S; f)$, where $\eta$ is the learning rate. We notice that the batch size $|S|$ cannot be very small. Otherwise, the update can be too noisy due to the data variance or some types of SSL whose benefit replies on large batch sizes, such as contrastive learning [9, 23], would lose this benefit. However, online (generalized) label shift problem assumes we only have a small batch of unlabeled samples $S_t$ at time $t$; the batch size is set as 1 or 10 in the experiment of literature. Therefore, for the effectiveness of SSL techniques, instead of updating the feature extractor at each time step by $S_t$, we opt to accumulate the sample batch $S_{\tau c} \cup \cdots \cup S_{\tau(c+1)}$ and update the feature extractor once every $\tau$ time steps, and we call it *batch accumulation*.

Besides the help of effectiveness, *batch accumulation* also helps with time efficiency. The existing OLS methods only involve the updates for the small reweighting vector or the linear layer and the time cost is low. However, updating the feature extractor and re-training the linear layer on the full training set, which are introduced in Principle 2, are much computationally heavier. For example, when we evaluated the methods in the experiment with ResNet18 on CIFAR10 dataset, one step of FLHFTL only took 0.069 second, while one step of FLHFTL together with feature extractor update and linear layer re-training took 17.1 seconds, which is 247 times. After applying *batch accumulation* and we select $\tau = 100$, the additional time costs will be reduced by $(\tau - 1)/\tau = 99\%$.

*Batch accumulation* answers $Q_3$ – *we should update the feature extractor every $\tau$ steps for effectiveness and time efficiency*. We will study how *batch accumulation* helps in the experiments.

---

**Algorithm 1** Online label shift adaptation with online feature updates (OLS-OFU).

---

**Require:** An online label shift adaptation algorithm OLS, a self-supervised learning loss $\ell_{\mathrm{ssl}}$. A pre-trained model $f_0$ and initialize $f_1 = f_1'' := f_0$.

**for** $t = 1, \cdots, T$ **do**

    **Input at time** $t$**:** Samples $S_1 \cup \cdots \cup S_t$, models $\{f_1, \cdots, f_t\}$, train set $D_0$, validation set $D_0'$.

    1. Run the revised version of OLS, that is, OLS-R, and get $f_{t+1}'$.

    **If** $t\%\tau \neq 0$, $f_{t+1} := f_{t+1}'$, $f_{t+1}'' := f_t''$ **Else**, go to the next step 2-3.

        2. Update the feature extractor $\theta_t^{\mathrm{feat}}$ in $f_{t+1}'$ by Equation 4. Replace $\theta_t^{\mathrm{feat}}$ in $f_{t+1}'$ by $\theta_{t+1}^{\mathrm{feat}}$.

        3. Re-train the last linear layer by Equation 5, calibrate the model and get $f_{t+1}''$.

    **Output at time** $t$**:** For the reweighting OLS methods, denote the latest reweighting vector from Step 1 is $p_{t+1}$ and we define $f_{t+1} := g(\cdot; f_{t+1}'', p_{t+1})$; else, we define $f_{t+1} := f_{t+1}'$.

**end for**

---

## 3.2 Online Label Shift Adaptation with Online Feature Updates

We have discussed three principles and now we can finalize our algorithm *Online Label Shift adaptation with Online Feature Updates* (OLS-OFU; Algorithm 1), which requires a self-supervised learning loss $\ell_{\mathrm{ssl}}$ and one of existing OLS methods in the literature, which either reweights the offline pre-trained model $f_0$ or updates the last linear layer. In the training stage, we train $f_0$ by minimizing the supervised and self-supervised loss together defined on train data. In the test stage, OLS-OFU comprises three steps at each time step $t$: (1) running the refined version of OLS, which we refer to as OLS-R, (2) updating the feature extractor, and (3) re-training the last linear layer. As suggested by Principle 3 in Section 3.1, steps (2-3) only run every $\tau$ steps. We illustrate the details of these three steps are elaborated below.

**(1) Running the Revised OLS.** At the beginning of the time $t$, we use $f_t''$ to denote the model within the latest feature extractor and the re-trained linear model (using data $\{S_0, \cdots, S_{t-1}\}$). The high level idea of our framework centers on substituting the pre-trained model $f_0$ used in existing OLS methods with our updated model $f_t''$, which we call *OLS-R*. To provide an overview, we examine common OLS methods (FLHFTL, FTH, ROGD, UOGD, and ATLAS) and identify two primary use cases of $f_0$. Firstly, all OLS methods rely on an unbiased estimator $s_t$ of the label distribution $q_t$ and $f_0$ is a part of the estimator. Secondly, the hypothesis $f$ is some weights of reweighting the $f_0$ output or only the last linear layer in $f_0$ is updated. We illustrate all revised OLS methods formally in Appendix C. We use $f_{t+1}'$ to denote the model after running the OLS-R.

**(2) Updating the Feature Extractor.** As guided by Principle 1 in Section 3.1, the step of updating the feature extractor should be after the Revised-OLS module (step (1)) for theoretical guarantees. We now introduce how to utilize the SSL loss $\ell_{\mathrm{ssl}}$ to update the feature extractor based on the accumulated batch of unlabeled test data $S_{\tau c} \cup \cdots \cup S_{\tau(c+1)}$ at timestep $t = \tau(c+1)$. Specifically, let $\theta_t^{\mathrm{feat}}$ denote the parameters of the feature extractor in $f_{t+1}'$. The update of $\theta_{t+1}^{\mathrm{feat}}$ is given by:

$$\theta_{t+1}^{\mathrm{feat}} := \theta_t^{\mathrm{feat}} - \eta \cdot \nabla_{\theta^{\mathrm{feat}}} \ell_{\mathrm{ssl}}(S_{\tau c} \cup \cdots \cup S_{\tau(c+1)}; f_{t+1}'). \tag{4}$$

Then we update the feature extractor in $f_{t+1}'$ with $\theta_{t+1}^{\mathrm{feat}}$.

**(3) Re-training Last Linear Layer.** Principle 2 in Section 3.1 suggests this step of re-training the last linear layer on the training distribution. The re-training starts from random initialization while the feature extractor remains frozen. The training objective of $\theta_{t+1}^{\mathrm{linear}}$ is to minimize the average cross-entropy loss over train data $D_0$. We denote the model with the frozen feature extractor $\theta_{t+1}^{\mathrm{feat}}$ as $f(\cdot|\theta_{t+1}^{\mathrm{feat}}, \theta^{\mathrm{linear}})$. The objective for re-training the last linear layer can be written as follows:

$$\theta_{t+1}^{\mathrm{linear}} := \arg\min_{\theta_{\mathrm{linear}}} \sum_{(x,y) \in D_0} \ell_{\mathrm{ce}}\left(f(x|\theta_{t+1}^{\mathrm{feat}}, \theta^{\mathrm{linear}}), y\right). \tag{5}$$

We calibrate the model $f(\cdot|\theta_{t+1}^{\mathrm{feat}}, \theta_{t+1}^{\mathrm{linear}})$ by temperature calibration [21] using the validation set $D_0'$ and denote the model after calibration as $f_{t+1}''$.

**Output at time** $t$**.** Finally, we define $f_{t+1}$ for the next time step prediction. For the reweighting OLS methods (ROGD, FTH, FLHFTL), denote the latest reweighting vector from Step 1 as $p_{t+1}$ and define $f_{t+1} := g(\cdot; f_{t+1}'', p_{t+1})$. For those which optimize the last linear layer (UOGD, ATLAS), we

define $f_{t+1} := f'_{t+1}$ – as stated in Principle 2 in Section 3.2, the $f''_t$ including retrained last linear layer serves specially for the reweighting OLS method.

**The requirement of the training data.** Notice that in step (3), we need the training data to retrain the linear classifier, which potentially brings the additional cost of memory and computation in the test time. As discussed in Principle 2 and lines 248-252, retraining the last linear layer is designed only for three previous OLS methods (ROGD, FTH, FLHFTL); our algorithm for two other OLS methods (UOGD and ATLAS) in the literature are independent of this step. As for ROGD, FTH, or FLHFTL, we interestingly find that OLS-OFU without this re-training step, which means that we update the feature extractor but reuse the pretrained linear classifier, still has a certain advantage over the original OLS methods – although more stored training data results in the more significant benefit of OLS-OFU. This suggests that in practice, if we reduce the amount of stored training data due to the memory or computational constraint, OLS-OFU is still effective.

### 3.3 Performance Guarantee for Online Label Shift Adaptation

One main advantage of the original OLS methods is that they exhibit theoretical guarantees in terms of regret convergence for online label shift settings. Next, we will show how OLS-OFU demonstrates analogous theoretical guarantees with the incorporation of additional online feature updates. Due to limited space, we illustrate the theoretical results pertaining to FLHFTL-OFU here, and present the results for ROGD-OFU, FTH-OFU, UOGD-OFU, and ATLAS-OFU in Appendix D.

**Theorem 1.** *[Regret convergence for FLHFTL-OFU] Suppose we choose the OLS-R to be FLHFTL-R (Algorithm 6) from Baby et al. [6]. Let $f_t^{\mathrm{flhftl-ofu}}$ be the output at time step $t-1$ from Algorithm 1, that is $g(\cdot; f''_t, \frac{\tilde{q}_t}{q_0})$. Under Assumptions 1 and 2 in Baby et al. [6], FLHFTL-OFU has the guarantee:*

$$\mathbb{E}\left[\frac{1}{T}\sum_{t=1}^{T}\ell(f_t^{\mathrm{flhftl-ofu}}; \mathcal{P}_t^{\mathrm{test}})\right] \leq \mathbb{E}\left[\frac{1}{T}\sum_{t=1}^{T}\ell(g(\cdot; f''_t, \frac{q_t}{q_0}); \mathcal{P}_t^{\mathrm{test}})\right] + O\left(\frac{K^{1/6}V_T^{1/3}}{T^{1/3}} + \frac{K}{\sqrt{T}}\right),$$
(6)

*where $V_T := \sum_{t=1}^{T}\|q_t - q_{t-1}\|_1$, $K$ is the number of classes, and the expectation is taken w.r.t. randomness in the revealed co-variates. This result is attained without prior knowledge of $V_T$.*

**How do online feature updates contribute to the bound?** We observe that the upper bound of the test loss has two terms. The first term is the loss of the model within the up-to-date feature extractor when the knowledge of label distribution $q_t$ is known. Any improvement from SSL could be reflected in this first term. The second term is to quantify the loss gap between the knowledge of label distribution $q_t$ and the estimation of the label distribution by the online learning technique.

**When do online feature updates improve the guarantee from FLHFTL to FLHFTL-OFU?** If we do not make any update to the feature extractor (i.e. $f''_t = f_0, \forall t$), the upper bound in Equation 6 would be naturally reduced to the theoretical guarantee of FLHFTL [6]. Moreover, in the following event of $f''_t$ ($t \in [T]$):

$$\mathbb{E}\left[\frac{1}{T}\sum_{t=1}^{T}\ell(g(\cdot; f''_t, \frac{q_t}{q_0}); \mathcal{P}_t^{\mathrm{test}})\right] < \frac{1}{T}\sum_{t=1}^{T}\ell(g(\cdot; f_0, \frac{q_t}{q_0}); \mathcal{P}_t^{\mathrm{test}}),$$
(7)

our theorem will guarantee that the loss of FLHFTL-OFU converges to a smaller value than the one of the original FLHFTL, resulting in a better upper bound. We substantiate this improvement through empirical evaluation in Section 4.

### 3.4 Online Feature Updates Improve Online Generalized Label Shift Adaptation

The generalized label shift is harder because the feature map $h$ is unknown and naively applying OLS methods might raise the challenge due to the violation of the label shift assumption. Fortunately, existing research in test-time training (TTT) [46, 48, 36, 39] demonstrates that feature updates driven by SSL align the source and target domains in feature space. When the source and target domains achieve perfect alignment, such feature extractor effectively serves as the feature map $h$ as assumed in generalized label shift. Therefore, the sequence of feature extractors in $f_1, \cdots, f_T$ generated by Algorithm 1 progressively approximates the underlying $h$. This suggests that, compared to the original OLS, OLS-OFU experiences a milder violation of the label shift assumption within the feature space and is expected to have better performance in online generalized label shift settings.

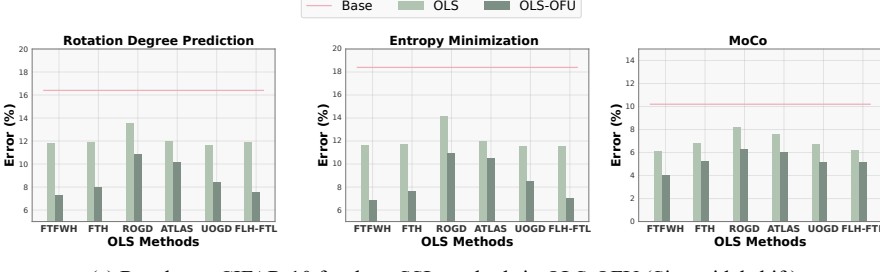

(a) Results on CIFAR-10 for three SSL methods in OLS-OFU (Sinusoidal shift)

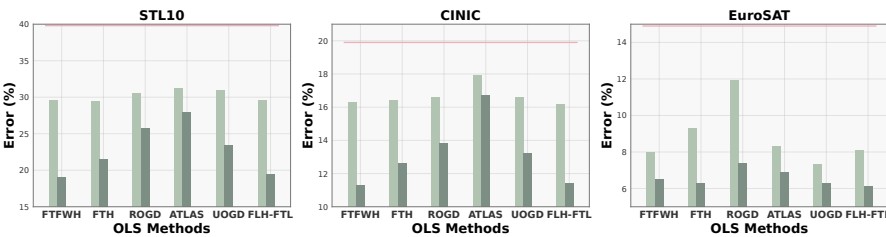

(b) Results on three more datasets (rotation degree prediction, Sinusoidal shift)

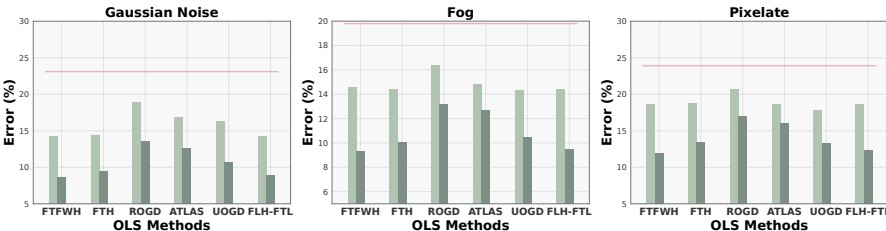

(c) Results on three types of corruptions in CIFAR-10C (rotation degree prediction, Sinusoidal shift)

Figure 2: Evaluation of OLS and OLS-OFU.

|  | FTFWH | FTH | ROGD | ATLAS | UOGD | FLH-FTL |
|---|---|---|---|---|---|---|
| OLS | 11.9% / 0.441 | 12.04% / 0.705 | 13.65% / 0.937 | 12.18% / 43.2 | 11.54% / 297 | 12.02% / 1.15 |
| OLS-OFU ($\tau = 1$) | 11.3% / 283 | 11.2% / 286 | 13.9% / 385 | 11.6% / 47 | 11.4% / 319 | 11.2% / 285 |
| OLS-OFU ($\tau = 10$) | 9.93% / 28.8 | 10.2% / 29.6 | 13.1% / 41.9 | 11.6% / 45.1 | 10.5% / 306 | 9.93% / 29.9 |
| OLS-OFU ($\tau = 50$) | 8.15% / 6.47 | 8.89% / 6.91 | 12% / 9.49 | 11.3% / 44.8 | 9.27% / 303 | 8.27% / 7.08 |
| OLS-OFU ($\tau = 100$) | 7.33% / 3.57 | 8% / 3.85 | 10.8% / 5.29 | 10.1% / 44.2 | 8.35% / 303 | 7.45% / 4.15 |
| OLS-OFU ($\tau = 500$) | 11.6% / 1.27 | 11.7% / 1.53 | 13.5% / 2.09 | 12.1% / 43.6 | 11.3% / 298 | 11.7% / 1.94 |

Table 1: Average error/time (minutes) of 6 original OLS methods versus OLS-OFU with various frequency $\tau$ in *batch accumulation*. The SSL in OLS-OFU is the rotation degree prediction.

## 4 Experiment

In this section, we initiate OLS-OFU with three popular SSL techniques and empirically evaluate how OLS-OFU improves the original OLS methods on both online label shift and online generalized label shift on various datasets and shift patterns[3].

### 4.1 Experiment Set-up

**Dataset and Label Shift Settings.** For online label shift, we evaluate the efficacy of our algorithm on CIFAR-10 [29], STL10 [12], CINIC [13], and EuroSAT [25]. For each dataset, we split the original train set into the offline train (i.e., $D_0$) and validation sets (i.e., $D_0'$) following a ratio of $4 : 1$. At the online test stage, unlabeled batches are sampled from the test set. For online generalized label shift, the offline train and validation sets are the CIFAR-10 images. The test unlabeled batches are drawn from CIFAR-10C [26], a benchmark with the same objects as CIFAR-10 but with various types of corruption. We experiment with three types of corruptions with CIFAR-10C: Gaussian noise, Fog,

---
[3]Code is released at https://github.com/dattasiddhartha/online-feature-updates-olsofu

|  | FTFWH | FTH | ROGD | ATLAS | UOGD | FLHFTL |
|---|---|---|---|---|---|---|
| OLS-OFU($\tau = 1$) | 11.3% | 11.2% | 13.9% | 11.6% | 11.4% | 11.2% |
| OLS-OFU-difforder($\tau = 1$) | 12.33% | 12.12% | 14.35% | 12.10% | 11.91% | 12.08% |
| OLS-OFU($\tau = 100$) | 7.23% | 7.91% | 10.81% | 10.12% | 8.33% | 7.53% |
| OLS-OFU-difforder($\tau = 100$) | 8.02% | 8.63% | 11.19% | 10.55% | 8.80% | 8.29% |

Table 2: Ablation study on the order between OLS and the feature update step in OLS-OFU.

and Pixelate. We follow Bai et al. [7] and Baby et al. [6] to simulate the online label distribution shift with two shift patterns: Sinusoidal shift and Bernoulli shift. We experiment with $T = 1000$ and batch size $B = 10$ at each time step, following Baby et al. [6]. See more details of dataset set-up and online shift patterns in Appendix E.1.

**Evaluation Metric.** We report the average error during test, i.e., $\frac{1}{TB} \sum_{t=1}^{T} \sum_{x_t \in S_t} \mathbb{1}\left(f_t(x_t) \neq y_t\right)$, where $(x_t, y_t) \sim \mathcal{P}_t^{\text{test}}$, to approximate $\frac{1}{T} \sum_{t=1}^{T} \ell(f_t; \mathcal{P}_t^{\text{test}})$ for the evaluation efficiency.

**Self-supervised learning methods in OLS-OFU.** In the experiment, we narrow our focus on three particular SSL techniques in the evaluation for classification tasks: *rotation degree prediction* [16, 46], *entropy minimization* [18, 48] and *MoCo* [23, 10, 11]. It is important to note that this concept extends beyond these three SSL techniques, and the incorporation of more advanced SSL techniques to further elevate the performance. Appendix E.1 gives more details of these SSL techniques.

**Set-ups of OLS methods, OLS-OFU and baselines.** We perform an extensive evaluation of 6 OLS methods in the literature: FTFWH, FTH, ROGD, UOGD, ATLAS, and FLHFTL by following the implementation in Baby et al. [6]. We report the performance of our method OLS-OFU (Algorithm 1) applied on top of each OLS and 3 SSL methods introduced above. The frequency parameter $\tau$ is fixed as 100 for most experiments unless we particularly mention it. Additionally, by following the setup in [49, 6], we report one baseline score *Base*, which uses the fixed pre-trained model $f_0$ to predict the labels at all time steps.

## 4.2 Results

**Main Results: comparison between OLS-OFU and OLS under online (generalized) label shift.** Figure 2(a) shows the performance comparison between OLS-OFU, implemented with *three SSL methods*, and their corresponding OLS counterparts on CIFAR-10 under the scenario of *classical online label shift*. Figure 2(b) shows the results on *three more datasets* with SSL technique in OLS-OFU being rotation degree prediction. Figure 2(c) shows the results on CIFAR-10C datasets for evaluating methods on *online generalized label shift*. The online shift pattern in Figure 2 is the sinusoidal shift and similar results on the Bernoulli shift are in Appendix E.2. We have two main observations from the results. First, we find our OLS-OFU method achieves substantial improvements over existing OLS methods, which is *as significant as to the gains existing OLS methods have over the baseline (i.e., without distribution shift adaptations)*. This demonstrates that integrating online feature updates is as effective in solving online distribution shifts as the fundamental online label shift method itself. Second, the improvement is consistent across all six original OLS methods and three SSL techniques on all datasets, which further demonstrates our OLS-OFU is general enough to incorporate future OLS methods and more advanced SSL techniques as well.

**Validating Principle 1: ablation study on the order between OLS and the feature update step in OLS-OFU.** We compare OLS-OFU with its other variant named OLS-OFU-difforder where we update SSL first and run OLS later (which violates Principle 1). We compare these two algorithms across all previous 6 OLS methods and two choices of batch accumulation $\tau = 1$ and $\tau = 100$. The dataset is CIFAR-10, the SSL is rotation degree prediction and the shift pattern is the sinusoidal shift. We report the average error every $\tau$ time step when the feature extractor is updated, so the order matters. From Table 2, we can observe that benefiting from Principle 1, the error OLS-OFU is consistently lower than OLS-OFU-difforder. This means that Principle 1 indeed is not only necessary for the correctness of theoretical guarantee but also plays an important role in practice.

**Validating Principle 3: ablation study on frequency $\tau$ in *batch accumulation* in OLS-OFU.** In section 3.1, we introduce the *batch accumulation* to flavor the effectiveness of the SSL and reduce the additional time cost of OLS-OFU compared with the original OLS methods. In Table 1, we

| % Stored Training Data | 100% | 80% | 60% | 40% | 20% | 10% | 5% | 0% | OLS only |
|---|---|---|---|---|---|---|---|---|---|
| **FTH-OFU** | 8% | 8.18% | 8.68% | 9.49% | 9.54% | 9.67% | 9.81% | 10.24% | 12.04% |
| **ROGD-OFU** | 10.8% | 10.93% | 11.84% | 12.50% | 12.63% | 12.82% | 12.94% | 13.50% | 13.65% |
| **FLHFTL-OFU** | 7.45% | 7.62% | 8.04% | 8.91% | 9.51% | 10.11% | 10.34% | 10.43% | 12.02% |
| **FTFWH-OFU** | 7.33% | 7.48% | 7.92% | 8.40% | 8.91% | 9.86% | 10.03% | 10.41% | 11.9% |

Table 3: The effectiveness of OLS-OFU versus the percentage of stored training data in step (3).

evaluate OLS-OFU on CIFAR-10 with various $\tau$ while initiating OLS-OFU with rotation degree prediction and compare the average error and time with OLS. We have these observations across all OLS (columns): 1. OLS-OFU with all $\tau$ outperforms OLS; 2. The average error of OLS-OFU decreases from $\tau = 1$ (*w/o* batch accumulation) to $\tau = 100$ and starts to increase after and this is because the effectiveness of SSL benefits from $\tau > 1$ while larger $\tau$ means less updates and hence hurts the long term performance; 3. the additional time cost of OLS-OFU with larger $\tau$ is smaller and $\tau = 100$ gives a great balance of performance and time cost for all OLS (and SSL; the similar tables for other two SSL are presented in the Appendix E.3). From the results, we recommend $\tau = 100$ as a good starting point for choosing this parameter.

**Ablation study on the amount of stored training data in step (3).** We further study how the amount of training data stored influences the effectiveness of our OLS-OFU when the OLS is ROGD, FTH, or FLHFTL, which is dependent on step (3). We experiment with the CIFAR-10 dataset and the shift pattern of sinusoidal shift; the SSL is chosen as rotation degree prediction. The percentage of stored training data (out of the whole training set of size 10,000) is varied in $\{100\%, 80\%, 60\%, 40\%, 20\%, 10\%, 5\%, 0\%\}$; 0% means that we still update the feature extractor but reuse the pretrained linear classifier. The results are reported in Table 3. We can observe that with less stored training data for retraining the last linear layer, the error of OLS-OFU would increase gradually. However, an important finding is that even with 0% stored training data, the error of OLS-OFU is still lower than the OLS without feature extractor updates. This can actually explained by the original test-time training papers [46, 48, 36, 39], where they only update the feature extractor without refining the last linear layer and still have substantial benefit. The results suggest that a large amount of stored training data is not necessary for the effectiveness of OLS-OFU, while more stored data can bring more benefits.

**Empirical validation of Equation 7.** In Section 3.3, we argued that when the inequality in Equation 7 holds, the loss of FLHFTL-OFU exhibits a tighter upper bound compared to FLHFTL. Figure 3 presents the RHS (corresponds to OLS) and LHS (corresponds to OLS-OFU with SSL loss as rotation degree prediction) of Equation 7. We perform the study over eight different settings and vary the domain shift and online shift patterns. It is evident that OLS-OFU yields improvements on the *baseline* of the regret as shown in Equation 7. Appendix E.4 validates this inequality for other SSL.

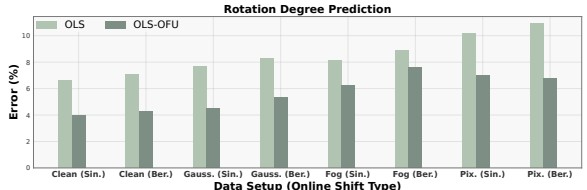

Figure 3: Empirical validation of Equation 7. Clean denotes the experiment on CIFAR-10, and others denote the corruption type. They are paired with two online shift patterns: Sinusoidal and Bernoulli.

## 5 Conclusion and Future Work

We focus on online (generalized) label shift adaptation and present a novel framework OLS-OFU, which harnesses the power of self-supervised learning to enhance feature representations dynamically during testing, leading to improved predictive models and better test time performance.

**Discussion and future work.** One promising direction is to extend the idea of this paper to online covariate shift, for example, the algorithm in Zhang et al. [53] freezes the feature extractor and only updates the linear layer. Another possible direction is to consider a more realistic domain shift scenario within the generalized label shift setting — domain shift types may vary over time or be even more challenging, such as shifting from cartoon images to realistic images.

## Acknowledgments and Disclosure of Funding

RW and KQW are supported by grants from LinkedIn, DARPA Geometries of Learning, and the National Science Foundation NSF (1934714). RW is also supported by grants from the National Science Foundation NSF (CIF-2402817, CNS-1804829), SaTC-2241100, CCF-2217058, ARO-MURI (W911NF2110317), and ONR under N00014-24-1-2304. YW and DB are partially supported by NSF Award #2134214.

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

## A  Further Related Work

**Offline distribution shift and domain shift.** Offline label shift and covariate shift have been studied for many years. Some early work [42, 33] assumes the knowledge of how the distribution is shifted. Later work [45, 51, 28, 19, 35, 1, 4, 14] relaxes this assumption and estimates this knowledge from unlabeled test data. A recent work by [15] considers a relaxed version of offline label shift problem where the class-conditionals can change between train and test domains in a restrictive way. Extending their results to the case of relaxed online version of generalized label shift is an interesting future direction.

**Online distribution shift with provable guarantees.** There has been several work modeling online distribution shift as the classic online learning problem [49, 7, 6, 53], which leverage the classical online learning algorithms [44, 8, 5] to bound the static or dynamic regret. Qian et al. [40] focuses on a special setting in online label shift, where there can be new classes occurring in the test stage. They handle the new class by unsupervised estimating the portion of unseen data. However, none of them updates the feature extractor in a deep learning model but only the last linear layer or the post-hoc linear reweighting vectors. Our proposed method OLS-OFU utilizes the deep learning SSL to improve the feature extractor, which brings better performance.

**Domain shift adaptation within online streaming data.** When we consider the most authentic online learning setup where the learner only receives the unlabeled samples, the most representative idea is test-time training [46, 48, 36, 39], which utilizes a (deep learning) self-supervised loss to online update the model. However, it focuses on how to adapt to a *fixed* domain shifted distribution from online streaming data and is not designed for how to adapt to continuous distribution changes during the test stage, while our algorithm concentrates the later problem. Besides test-time training, Hoffman et al. [27] and [38] study the online domain shift for specific visual applications.

## B  Proof of Proposition 1

*The proof of Proposition 1.* We first write the derivation that $\sum_{y \in \mathcal{Y}} s_t[y] \cdot \nabla_f \mathbb{E}_{x \sim \mathcal{P}^{\text{train}}(\cdot|y)} \ell_{\sup}(f_t(x), y)$ is an unbiased estimator of $\nabla_f \ell(f_t; \mathcal{P})$ when $f_t$ is indepen-

---

**Algorithm 2** Revised ROGD for online feature updates ROGD-R. See the original version in Equation 7 and Equation 8 in [49].

---

**Require:** Learning rate $\eta$.
**for** $t = 1, \cdots, T$ **do**
    **Input at time** $t$**:** Samples $S_1 \cup \cdots \cup S_t$, models $\{f_1, \cdots, f_t\}$, and intermediate model $\{f_1'', \cdots, f_t''\}$ from step 3 in Algorithm 1, the validation set $D_0'$, the training label marginal $q_0 := \mathcal{P}^{\text{train}}(y)$.
    1. Compute the unbiased estimator for label marginal distribution:
        $s_t = \frac{1}{|S_t|} \sum_{x_t \in S_t} C_{f_t'', D_0'}^{-1} f_t''(x_t).$        ▷ In the original ROGD, it is $f_0$ rather than $f_t''$.
    2. Grab the weight $p_t$ from $f_t$.
    3. Update $p_{t+1} := \text{Proj}_{\Delta^{K-1}} \left[ p_t - \eta \cdot J_p(p_t)^\top s_t \right]$,
        where $J_{p,f_t''}(p_t) = \frac{\partial}{\partial p} (1 - \text{diag}(C_{f_t'', D_0, p}))|_{p=p_t}$, and let $f_{t+1}$ be a reweighting version of
        $f_t''$ by the weight $\left( \frac{p_{t+1}[k]}{q_0[k]} : k = 1, \cdots K \right)$    ▷ In the original ROGD, it is $f_0$ rather than $f_t''$.
    **Output at time** $t$**:** $f_{t+1}'$.
**end for**

---

dent of $S_t$.

$$\mathbb{E}_{S_t}\left[\sum_{y\in\mathcal{Y}} s_t[y]\cdot\nabla_f\mathbb{E}_{x\sim\mathcal{P}^{\mathrm{train}}(\cdot|y)}\ell_{\mathrm{sup}}(f_t(x),y)\right] = \sum_{y\in\mathcal{Y}}\mathbb{E}_{S_t}\left[s_t[y]\cdot\nabla_f\mathbb{E}_{x\sim\mathcal{P}^{\mathrm{train}}(\cdot|y)}\ell_{\mathrm{sup}}(f_t(x),y)\right]$$

$$= \sum_{y\in\mathcal{Y}}\mathbb{E}_{S_t}\left[s_t[y]\right]\cdot\nabla_f\mathbb{E}_{x\sim\mathcal{P}^{\mathrm{train}}(\cdot|y)}\ell_{\mathrm{sup}}(f_t(x),y)$$

$$= \sum_{y\in\mathcal{Y}} q_t[y]\cdot\nabla_f\mathbb{E}_{x\sim\mathcal{P}^{\mathrm{train}}(\cdot|y)}\ell_{\mathrm{sup}}(f_t(x),y)$$

$$= \sum_{y\in\mathcal{Y}} q_t[y]\cdot\nabla_f\mathbb{E}_{x\sim\mathcal{P}_t^{\mathrm{test}}(\cdot|y)}\ell_{\mathrm{sup}}(f_t(x),y)$$

$$= \nabla_f\ell(f_t;\mathcal{P}).$$

The third equality is as how $s_t$ is constructed. The fourth equality holds by the label shift assumption. We can check the second equality: if $f_t$ is dependent of $S_t$, the correctness of the second equality is not guaranteed and so is the overall unbiased property derivation.

$\square$

## C  The Revision for Previous Online Label Shift Adaptation Algorithms

The revised algorithms to be used in the main algorithm OLS-OFU (Algorithm 1) are FTH-R (Algorithm 3), UOGD-R (Algorithm 4), ROGD-R (Algorithm 2), ATLAS-R (Algorithm 5), FLHFTL-R (Algorithm 6).

## D  Theorems for OLS and Proofs

In this section, we present the theoretical results of FLHFTL-OFU, ROGD-OFU, FTH-OFU, UOGD-OFU, ATLAS-OFU and their proofs. The proofs are mostly the same as the proofs for the original algorithms with small adjustments. As our results are not straight corollaries for the original theorems, we write the full proofs here for the completeness.

### D.1  Theorem for FLHFTL-OFU

Before proving Theorem 2 (in Section 3.3 ) we recall the assumption from Baby et al. [6] for convenience. We refer the reader to Baby et al. [6] for justifications and further details of the assumptions.

**Assumption 1.** *Assume access to the true label marginals $q_0\in\Delta_K$ of the offline train data and the true confusion matrix $C\in\mathbb{R}^{K\times K}$. Further the minimum singular value $\sigma_{min}(C)=\Omega(1)$ is bounded away from zero.*

---

**Algorithm 3** Revised FTH for online feature updates (FTH-R). See the original version in Equation 9 in [49].

---
**for** $t=1,\cdots,T$ **do**

  **Input at time** $t$**:** Samples $S_1\cup\cdots\cup S_t$, models $\{f_1,\cdots,f_t\}$, and intermediate model $\{f_1'',\cdots,f_t''\}$ from step 3 in Algorithm 1, the validation set $D_0'$, the train label marginal $q_0:=\mathcal{P}^{\mathrm{train}}(y)$.

  1. Compute the unbiased estimator for label marginal distribution:

   $s_t=\frac{1}{|S_t|}\sum_{x_t\in S_t}C_{f_t,D_0'}^{-1}f_t''(x_t)$.                 ▷ In the original FTL, it is $f_0$ rather than $f_t''$.

  2. Compute $p_{t+1}=\frac{1}{t}\sum_{\tau=1}^t s_\tau$

  3. Let $f_{t+1}$ be a reweighting version of $f_t''$ by

   the weight $\left(\frac{p_{t+1}[k]}{q_0[k]}:k=1,\cdots K\right)$                 ▷ In the original FTL, it is $f_0$ rather than $f_t''$.

  **Output at time** $t$**:** $f_{t+1}'$.

**end for**

---

**Algorithm 4** Revised UOGD for online feature updates (UOGD-R). See the original version in Equation 9 in [7].

---

**Require:** The learning rate $\eta$.
**for** $t = 1, \cdots, T$ **do**

    **Input at time** $t$**:** Samples $S_1 \cup \cdots \cup S_t$, models $\{f_1, \cdots, f_t\}$, and intermediate model $\{f_1'', \cdots, f_t''\}$ from step 3 in Algorithm 1, the validation set $D_0'$, the train label marginal $q_0 := \mathcal{P}^{\text{train}}(y)$.

    1. Compute the unbiased estimator for label marginal distribution:

        $s_t = \frac{1}{|S_t|} \sum_{x_t \in S_t} C^{-1}_{f_t'', D_0'} f_t''(x_t).$             ▷ In the original UOGD, it is $f_0$ rather than $f_t''$.

    2. Grab the weight $w_t$ from the last linear layer of $f_t$.

    3. Update $w_{t+1} := w_t - \eta \cdot \frac{\partial}{\partial w} J_w(w_t)^\top s_t$, where $J_w(w_t) = \frac{\partial}{\partial w}(\hat{R}_t^1(w), \cdots, \hat{R}_t^K(w))|_{w=w_t}$,
        $\hat{R}_t^k(w) = \frac{1}{|D_0^k|} \sum_{(x,y) \in D_0^k} \ell_{\text{ce}}(f(x|\theta_t^{\text{feat}}, \theta^{\text{linear}} = w), y)$, $D_0^k$ denotes the set of data with
        label $k$ in $D_0$.                    ▷ In the original UOGD, it is $\theta_0^{\text{feat}}$ rather than $\theta_t^{\text{feat}}$.

    4. Let $f_{t+1}$ be $f(\cdot|\theta_t^{\text{feat}}, w_{t+1})$.

    **Output at time** $t$**:** $f_{t+1}'$.

**end for**

---

**Algorithm 5** Revised ATLAS for online feature updates (ATLAS-R). See the original version in Equation 9 in [7].

---

**Require:** The learning rate pool $\mathcal{H}$ with size N; Meta learning rate $\varepsilon$; $\forall i \in [N], p_{1,i} = 1/N$ and $w_{1,i} = \theta_{f_0}^{\text{linear}}$.
**for** $t = 1, \cdots, T$ **do**

    **Input at time** $t$**:** Samples $S_1 \cup \cdots \cup S_t$, models $\{f_1, \cdots, f_t\}$, and intermediate model $\{f_1'', \cdots, f_t''\}$ from step 3 in Algorithm 1, the validation set $D_0'$, the train label marginal $q_0 := \mathcal{P}^{\text{train}}(y)$.

    1. Compute the unbiased estimator for label marginal distribution:

        $s_t = \frac{1}{|S_t|} \sum_{x_t \in S_t} C^{-1}_{f_t, D_0'} f_t''(x_t).$           ▷ In the original ATLAS, it is $f_0$ rather than $f_t''$.

    **for** $i \in [N]$ **do**

        2. Update $w_{t+1,i} := w_{t,i} - \eta_i \cdot \frac{\partial}{\partial w} J_w(w_{t,i})^\top s_t$, where
        $J_w(w_{t,i}) = \frac{\partial}{\partial w}(\hat{R}_t^1(w), \cdots, \hat{R}_t^K(w))|_{w=w_{t,i}}$,
        $\hat{R}_t^k(w) = \frac{1}{|D_0^k|} \sum_{(x,y) \in D_0^k} \ell_{\text{ce}}(f(x|\theta_t^{\text{feat}}, w), y)$, $D_0^k$ denotes the set of data
        with label $k$ in $D_0$.             ▷ In the original ATLAS, it is $\theta_0^{\text{feat}}$ rather than $\theta_t^{\text{feat}}$.

    **end for**

    3. Update weight $p_{t+1}$ according to $p_{p_{t,i}} \propto \exp(-\varepsilon \sum_{\tau=1}^{t-1} \hat{R}_\tau(\mathbf{w}_{\tau,i}))$

    3. Compute $w_{t+1} = \sum_{i=1}^N p_{t+1,i} w_{t+1,i}$. Let $f_{t+1}$ be $f(\cdot|\theta_t^{\text{feat}}, w_{t+1})$.

    **Output at time** $t$**:** $f_{t+1}'$.

**end for**

---

**Algorithm 6** Revised FLHFTL for online feature updates (FLHFTL-R); See the original version in [6].

---

**Require:** Online regression oracle ALG.
**for** $t = 1, \cdots, T$ **do**

    **Input at time** $t$**:** Samples $S_1 \cup \cdots \cup S_t$, models $\{f_1, \cdots, f_t\}$, intermediate models $\{f_1'', \cdots, f_t''\}$, the validation set $D_0'$, the train label marginal $q_0 := \mathcal{P}^{\text{train}}(y)$.

    1. Compute the unbiased estimator for label marginal distribution: $s_t = \frac{1}{|S_t|} \sum_{x_t \in S_t} C^{-1}_{f_t'', D_0'} f_t''(x_t)$ ▷ In the original FLHFTL, it is $f_0$ rather than $f_t''$.

    2. Compute $\tilde{q}_{t+1} := \text{ALG}(s_1, \cdots, s_t)$

    3. Let $f_{t+1}'$ be a reweighting version of $f_t''$ by the weight $\left(\frac{\tilde{q}_{t+1}[k]}{q_0[k]} : k = 1, \cdots K\right)$ ▷ In the original FLHFTL, it is $f_0$ rather than $f_t''$.

    **Output at time** $t$**:** $f_{t+1}'$.

**end for**

---

**Assumption 2** (Lipschitzness of loss functions). *Let $\mathcal{D}$ be a compact and convex domain. Let $r_t$ be any probabilistic classifier. Assume that $L_t(p) := E\left[\ell(g(\cdot; r_t, p/q_0)|x_t\right]$ is $G$ Lipschitz with $p \in \mathcal{D} \subseteq \Delta_K$, i.e, $L_t(p_1) - L_t(p_2) \leq G\|p_1 - p_2\|_2$ for any $p_1, p_2 \in \mathcal{D}$. The constant $G$ need not be known ahead of time.*

**Theorem 2.** *[**Regret convergence for FLHFTL-OFU**] Suppose we choose the OLS-R to be FLHFTL-R (Algorithm 6) from Baby et al. [6]. Let $f_t^{\text{flhftl}-\text{ofu}}$ be the output at time step $t-1$ from Algorithm 1, that is $g(\cdot; f_t'', \frac{\tilde{q}_t}{q_0})$. Let $\sigma$ be the smallest among the the minimum singular values of invertible confusion matrices $\{C_{f_1'', D_0'}, \cdots C_{f_T'', D_0'}\}$. Then under Assumptions 1 and 2 in Baby et al. [6], FLHFTL-OFU has the guarantee below:*

$$\mathbb{E}\left[\frac{1}{T}\sum_{t=1}^{T}\ell(f_t^{\text{flhftl}-\text{ofu}}; \mathcal{P}_t^{\text{test}})\right] \leq \mathbb{E}\left[\frac{1}{T}\sum_{t=1}^{T}\ell(g(\cdot; f_t'', \frac{q_t}{q_0}); \mathcal{P}_t^{\text{test}})\right] + O\left(\frac{K^{1/6}V_T^{1/3}}{\sigma^{2/3}T^{1/3}} + \frac{K}{\sigma\sqrt{T}}\right),$$
(8)

*where $V_T := \sum_{t=1}^{T}\|q_t - q_{t-1}\|_1$, $K$ is the number of classes, and the expectation is taken w.r.t. randomness in the revealed co-variates. This result is attained without prior knowledge of $V_T$.*

**Proof:** The proof follows the similar idea in the original online regression algorithm FLHFTL [22] and its variants for solving online label shift problem in Baby et al. [6].

The algorithm in Baby et al. [6] requires that the estimate $s_t$ in Line 1 of Algorithm 6 is unbiased estimate of the label marginal $q_t$. Since $f_t''$ in Algorithm 6 is independent of the sample $S_t$, and since we are working under the standard label shift assumption, due to Lipton et al. [35] we have that $C_{f_t'', D_0'}^{-1} \cdot \frac{1}{|S_t|}\sum_{x_t \in S_t} f_t''(x_t)$ forms an unbiased estimate of $E_{x \sim \mathcal{P}_t^{\text{test}}}[f_t''(x)]$. Further, from Lipton et al. [35], the reciprocal of standard deviation of this estimate is bounded below by minimum of the singular values of confusion matrices $\{C_{f_1'', D_0'}, \cdots C_{f_T'', D_0'}\}$.

Let $\tilde{q}_t$ be the estimate of the label marginal maintained by FLHFTL. By Lipschitzness, we have that

$$E[\ell(f_t^{\text{flhftl}-\text{ofu}}; \mathcal{P}_t^{\text{test}}) - \ell(g(\cdot; f_t'', p/q_0)] = E[L_t(\tilde{q}_t)] - E[L_t(q_t)]$$
(9)

$$\leq G \cdot E[\|\tilde{q}_t - q_t\|_2],$$
(10)

where the last line is via Assumption 2.

$$\sum_{t=1}^{T} E[\ell(f_t^{\text{flhftl}-\text{ofu}}; \mathcal{P}_t^{\text{test}}) - \ell(g(\cdot; f_t'', p/q_0)] \leq \sum_{t=1}^{T} G \cdot E[\|\tilde{q}_t - q_t\|_2]$$
(11)

$$\leq \sum_{t=1}^{T} G\sqrt{E\|\tilde{q}_t - q_t\|_2^2}$$
(12)

$$\leq G\sqrt{T\sum_{t=1}^{T}E[\|\tilde{q}_t - q_t\|_2^2]}$$
(13)

$$= \tilde{O}\left(K^{1/6}T^{2/3}V_T^{1/3}(1/\sigma_{min}^{2/3}(C)) + \sqrt{KT}/\sigma_{min}(C)\right),$$
(14)

where the second line is due to Jensen's inequality, third line by Cauchy-Schwartz and last line by Proposition 16 in Baby et al. [6]. This finishes the proof.

### D.2 Theorem for ROGD-OFU

We state the assumptions first for the later theorems. These assumptions are similar to Assumption 1-3 in [49].

**Assumption 3.** $\forall \mathcal{P} \in \{\mathcal{P}^{\text{train}}, \mathcal{P}_1^{\text{test}}, \cdots, \mathcal{P}_T^{\text{test}}\}$, $\text{diag}(C_{f,\mathcal{P}})$ *is differentiable with respect to $f$.*

**Assumption 4.** $\forall t \in [T]$, $\ell(g(\cdot; f_t'', p/q_0); \mathcal{P}_t^{\text{test}})$ *is convex in p, where $f_t''$ is defined in Algorithm 1.*

**Assumption 5.** $\sup_{p \in \Delta^{K-1}, i \in [K], t \in [T]} \|\nabla_p \ell(g(\cdot; f_t'', p/q_0); \mathcal{P}_t^{\text{test}})\|_2$ *is finite and bounded by L.*

**Theorem 3** (Regret convergence for ROGD-OFU). *If we run Algorithm 1 with ROGD-R (Algorithm 2) and $\eta = \sqrt{\frac{2}{T}\frac{1}{L}}$, under Assumption 3, 4, 5, ROGD-OFU satisfies the guarantee*

$$\mathbb{E}\left[\frac{1}{T}\sum_{t=1}^{T}\ell(f_t^{\text{ogd-ofu}}; \mathcal{P}_t^{\text{test}})\right] - \min_{p \in \Delta_K}\mathbb{E}\left[\frac{1}{T}\sum_{t=1}^{T}\ell(g(\cdot; f_t'', p/q_0); \mathcal{P}_t^{\text{test}})\right] \leq \sqrt{\frac{2}{T}}L. \quad (15)$$

$$\mathbb{E}\left[\frac{1}{T}\sum_{t=1}^{T}\ell(f_t^{\text{ogd}}; \mathcal{Q}_t)\right] - \min_{p \in \Delta_K}\mathbb{E}\left[\frac{1}{T}\sum_{t=1}^{T}\ell(g(\cdot; p, f_0, q_0); \mathcal{Q}_t)\right] \leq \sqrt{\frac{2}{T}}L. \quad (16)$$

**Proof:** The proof follows the similar idea in the original online leanring algorithm online gradient descent (OGD) [44] and the ROGD in Wu et al. [49]. For any fixed $p$,

$$\ell(f_t^{\text{rogd-ofu}}; \mathcal{P}_t^{\text{test}}) - \ell(g(\cdot; f_t'', p/q_0); \mathcal{P}_t^{\text{test}}) = \ell(g(\cdot; f_t'', p_t/q_0); \mathcal{P}_t^{\text{test}}) - \ell(g(\cdot; f_t'', p/q_0); \mathcal{P}_t^{\text{test}})$$
$$\leq (p_t - p) \cdot \nabla_p \ell(g(\cdot; f_t'', p_t/q_0); \mathcal{P}_t^{\text{test}})$$
$$= (p_t - p) \cdot J_{p,f_t''}(p_t)^\top \mathbb{E}_{S_t}[s_t | S_1, \cdots, S_{t-1}]$$
$$= \mathbb{E}_{S_t}[(p_t - p) \cdot J_{p,f_t''}(p_t)^\top s_t | S_1, \cdots, S_{t-1}],$$

where the last inequality holds by the fact that $(p_t - p) \cdot J_{p,f_t''}(p_t)^\top$ is independent of $\{S_1, \cdots, S_{t-1}\}$. To bound $(p_t - p) \cdot J_{p,f_t''}(p_t)^\top s_t$,

$$\|p_{t+1} - p\|_2^2 = \|\text{Prof}_{\Delta^{K-1}}(p_t - \eta \cdot J_{p,f_t''}(p_t)^\top s_t) - p\|_2^2$$
$$\leq \|p_t - \eta \cdot J_{p,f_t''}(p_t)^\top s_t - p\|_2^2$$
$$= \|p_t - p\|_2^2 + \eta^2 \|J_{p,f_t''}(p_t)^\top s_t\|_2^2 - 2\eta(p_t - p) \cdot (J_{p,f_t''}(p_t)^\top s_t).$$

This implies

$$(p_t - p) \cdot (J_{p,f_t''}(p_t)^\top s_t) \leq \frac{1}{2\eta}(\|p_t - p\|_2^2 - \|p_{t+1} - p\|_2^2) + \frac{\eta}{2}\|J_{p,f_t''}(p_t)^\top s_t\|_2^2$$

Thus

$$\mathbb{E}_{S_1, \cdots, S_T}\left[\frac{1}{T}\sum_{t=1}^{T}\ell(f_t^{\text{rogd-ofu}}; \mathcal{P}_t^{\text{test}}) - \frac{1}{T}\sum_{t=1}^{T}\ell(g(\cdot; f_t'', p/q_0); \mathcal{P}_t^{\text{test}})\right]$$
$$\leq \mathbb{E}_{S_1, \cdots, S_T}\left[\frac{1}{T}\sum_{t=1}^{T}\frac{1}{2\eta}(\|p_t - p\|_2^2 - \|p_{t+1} - p\|_2^2) + \frac{\eta}{2}\|J_{p,f_t''}(p_t)^\top s_t\|_2^2\right]$$
$$\leq \frac{1}{2\eta T}\|p_1 - p\|_2^2 + \frac{\eta}{2T}\sum_{t=1}^{T}\mathbb{E}_{S_1, \cdots, S_t}[\|J_{p,f_t''}(p_t)^\top s_t\|_2^2]$$
$$\leq \frac{1}{\eta T} + \frac{\eta L^2}{2} = \sqrt{\frac{2}{T}}L.$$

This bound holds for any p. Thus,

$$\mathbb{E}_{S_1, \cdots, S_T}\left[\frac{1}{T}\sum_{t=1}^{T}\ell(f_t^{\text{rogd-ofu}}; \mathcal{P}_t^{\text{test}})\right] - \min_{p \in \Delta^{K-1}}\mathbb{E}_{S_1, \cdots, S_T}\left[\frac{1}{T}\sum_{t=1}^{T}\ell(g(\cdot; f_t'', p/q_0); \mathcal{P}_t^{\text{test}})\right] \leq \sqrt{\frac{2}{T}}L.$$

### D.3 Theorem for FTH-OFU

We begin with two assumptions.

**Assumption 6.** *For any $\mathcal{P}^{\text{test}}$ s.t. $\mathcal{P}^{\text{test}}(x|y) = \mathcal{P}^{\text{train}}(x|y)$, denote $q_t := (\mathcal{P}_t^{\text{test}}(y = k) : k \in [K])$ and then*

$$\|q_t - \arg\min_{p \in \Delta^{K-1}}\ell(g(\cdot; f_t'', p/q_0); \mathcal{P}^{\text{test}})\| \leq \delta.$$

**Assumption 7.** $\forall \mathcal{P}^{\text{test}}$ *s.t.* $\mathcal{P}^{\text{test}}(x|y) = \mathcal{P}^{\text{train}}(x|y)$, $\sup_p \|\nabla_p \ell(g(\cdot; f_t'', p/q_0); \mathcal{P}^{\text{test}})\| \leq L$

**Theorem 4** (Regret convergence for FTH-OFU). *If we run Algorithm 1 with FTH-R (Algorithm 3) and assume $\sigma$ is no larger than the minimum singular value of invertible confusion matrices $\{C_{f_1'', D_0'}, \cdots C_{f_T'', D_0'}\}$, under Assumption 6 and 7 with $\delta = 0$, FTH-OFU satisfies the guarantee that with probability at least $1 - 2KT^{-7}$ over samples $S_1 \cup \cdots \cup S_T$,*

$$\frac{1}{T}\sum_{t=1}^{T} \ell(f_t^{\text{fth-ofu}}; \mathcal{P}_t^{\text{test}}) - \min_{p \in \Delta_K} \frac{1}{T}\sum_{t=1}^{T} \ell(g(\cdot; f_t'', p/q_0); \mathcal{P}_t^{\text{test}}) \leq O\left(\frac{\log T}{T} + \frac{1}{\sigma}\sqrt{\frac{K \log T}{T}}\right),$$
(17)

*where $K$ is the number of classes.*

**Proof:** The proof follows the similar idea in the original online leanring algorithm FTL [44] and the FTH in Wu et al. [49]. Denote $q_t := (\mathcal{P}_t^{\text{test}}(y = k) : k \in [K])$. By the Hoeffding and union bound, we have

$$\mathbb{P}\left(\forall t \leq T, \|p_{t+1} - \frac{1}{t}\sum_{\tau=1}^{t} q_\tau\| \leq \sqrt{K}\varepsilon_t\right) \geq 1 - \sum_{t=1}^{T} 2M \exp\left(-2\varepsilon_t^2 t/\sigma^2\right).$$

This implies that with probability at least $1 - \sum_{t=1}^{T} 2M \exp\left(-2\varepsilon_t^2 t/\sigma^2\right)$, $\forall p$,

$$\sum_{t=1}^{T} \ell(p_t; \mathcal{P}_t^{\text{test}}) - \sum_{t=1}^{T} \ell(g(\cdot; f_t'', p/q_0); \mathcal{P}_t^{\text{test}})$$

$$\leq \sum_{t=1}^{T} \ell(g(\cdot; f_t'', \frac{1}{t}\sum_{\tau=1}^{t} q_\tau/q_0); \mathcal{P}_t^{\text{test}}) - \sum_{t=1}^{T} \ell(g(\cdot; f_t'', p/q_0); \mathcal{P}_t^{\text{test}}) + L\sqrt{M} \cdot \sum_{t=1}^{T} \varepsilon_t$$

$$\leq \sum_{t=1}^{T} \ell(g(\cdot; f_t'', \frac{1}{t-1}\sum_{\tau=1}^{t-1} q_\tau/q_0); \mathcal{P}_t^{\text{test}}) - \sum_{t=1}^{T} \ell(g(\cdot; f_t'', \frac{1}{t}\sum_{\tau=1}^{t} q_\tau/q_0); \mathcal{P}_t^{\text{test}}) + L\sqrt{M} \cdot \sum_{t=1}^{T} \varepsilon_t$$

$$\leq \sum_{t=1}^{T} L \left\|\frac{1}{t-1}\sum_{\tau=1}^{t-1} q_\tau - \frac{1}{t}\sum_{\tau=1}^{t} q_\tau\right\| + L\sqrt{M} \cdot \sum_{t=1}^{T} \varepsilon_t$$

$$\leq \sum_{t=1}^{T} \frac{L}{t} \left\|\frac{1}{t-1}\sum_{\tau=1}^{t-1} q_\tau - q_t\right\| + L\sqrt{M} \cdot \sum_{t=1}^{T} \varepsilon_t$$

$$\leq \sum_{t=1}^{T} \frac{2L}{t} + L\sqrt{M} \cdot \sum_{t=1}^{T} \varepsilon_t.$$

If we take $\varepsilon_t = 2\sigma\sqrt{\frac{\ln T}{T}}$, the above is equivalent to: with probability at least $1 - 2KT^{-7}$

$$\frac{1}{T}\sum_{t=1}^{T} \ell(p_t; \mathcal{P}_t^{\text{test}}) - \min_p \frac{1}{T}\sum_{t=1}^{T} \ell(g(\cdot; f_t'', p/q_0); \mathcal{P}_t^{\text{test}}) \leq 2L\frac{\ln T}{T} + 4L\sigma\sqrt{\frac{K \ln T}{T}}$$

### D.4  Theorems for UOGD-OFU and ATLAS-OFU

**Theorem 5.** *[Regret convergence for UOGD-OFU] Let $f(\cdot; \theta_{f_t''}^{\text{feat}}, w)$ denote a network with the same feature extractor as that of $f_t''$ and a last linear layer with weight $w$. Let $f^{\text{uogd-ofu}} = f(\cdot; \theta_{f_t''}^{\text{feat}}, w_t)$, where $w_t$ is the weight maintained at round $t$ by Algorithm 4. If we run Algorithm 1 with UOGD in [7] and let step size be $\eta$, then under the same assumptions as Lemma 1 in [7], UOGD-OFU satisfies that*

$$\mathbb{E}\left[\frac{1}{T}\sum_{t=1}^{T} \ell(f^{\text{uogd-ofu}}; \mathcal{P}_t^{\text{test}}) - \frac{1}{T}\sum_{t=1}^{T} \min_{w \in \mathcal{W}} \ell(f(\cdot; \theta_{f_t''}^{\text{feat}}, w); \mathcal{P}_t^{\text{test}})\right] \leq O\left(\frac{K\eta}{\sigma^2} + \frac{1}{\eta T} + \sqrt{\frac{V_{T,\ell}}{T\eta}}\right),$$
(18)

*where $V_{T,\ell} := \sum_{t=2}^{T} \sup_{w \in \mathcal{W}} |\ell(f(\cdot; \theta_{f_t''}^{\text{feat}}, w); \mathcal{P}_t^{\text{test}}) - \ell(f(\cdot; \theta_{f_{t-1}''}^{\text{feat}}, w); \mathcal{P}_{t-1}^{\text{test}})|$, $\sigma$ denotes the minimum singular value of the invertible confusion matrices $\{C_{f_1'', D_0'}, \cdots C_{f_T'', D_0'}\}$ and $K$ is the number of classes and the expectation is taken with respect to randomness in the revealed co-variates.*

**Proof Sketch:** Recall that $\ell(f(\cdot;\theta^{\mathrm{feat}}_{f''_t}, w); \mathcal{P}^{\mathrm{test}}_t) := E_{(x,y)\sim\mathcal{P}^{\mathrm{test}}_t}\ell_{\mathrm{ce}}\left(f(x|\theta^{\mathrm{feat}}_{f''_t}, w), y\right)$.

This guarantee follows from the arguments in Bai et al. [7] from two basic facts below:

1. The risk $\ell(f(\cdot;\theta^{\mathrm{feat}}_{f''_t}, w); \mathcal{P}^{\mathrm{test}}_t)$ is convex in $w$ over a convex and compact domain $\mathcal{W}$.

2. It is possible to form unbiased estimates $\hat{G}_t(w) \in \mathbb{R}^K$ such that $E[\hat{G}_t(w)|S_{1:t-1}] = E_{((x,y)\sim\mathcal{P}^{\mathrm{test}}_t)}\nabla_w\ell_{\mathrm{ce}}\left(f(x|\theta^{\mathrm{feat}}_{f''_t}, w), y\right)$.

Hence we proceed to verify these two facts in our setup. Fact 1 is true because the cross-entropy loss is convex in any subset of the simplex and the last linear layer weights only defines an affine transformation which preserves convexity.

For fact 2, note that the $f''_t$ only uses the data until round $t-1$. So by the same arguments in Bai et al. [7], using the BBSE estimator defined from the classifier $f''_t$, the unbiased estimate of risk gradient can be defined.

Let $w_t$ be the weight of the last layer maintained by UOGD at round t. Let $u_{1:T}$ be any sequence in $\mathcal{W}$. Consequently we have for any round,

$$\ell(f^{\mathrm{uogd-ofu}}; \mathcal{P}^{\mathrm{test}}_t) - \ell(f(\cdot;\theta^{\mathrm{feat}}_{f''_t}, u_t)) = \ell(f(\cdot;\theta^{\mathrm{feat}}_{f''_t}, w_t) - \ell(f(\cdot;\theta^{\mathrm{feat}}_{f''_t}, u_t)) \tag{19}$$

$$\leq \langle \nabla_w\ell(f(\cdot;\theta^{\mathrm{feat}}_{f''_t}, w_t), w_t - u_t\rangle \tag{20}$$

$$= \langle E[\hat{G}_t(w_t)|S_{1:t-1}], w_t - u_t\rangle. \tag{21}$$

Rest of the proof is identical to Bai et al. [7].

Theorem 5 for UOGD-OLS shows a bound that depends on the learning rate and one can set up the learning carefully to get the optimal rate. However, this learning rate requires the prior information of $V_T$, which is typically unknown during the learning process. ATLAS-OLS overcomes this issue by applying the same online ensembling framework [54] as the original ATLAS. Without knowing this prior information, ATLAS-OLS is able to achieve the same rate of upper bound too, which is stated in the following theorem.

**Theorem 6** (Regret convergence for ATLAS-OFU). *Let $f(\cdot;\theta^{\mathrm{feat}}_{f''_t}, w)$ denote a network with the same feature extractor as that of $f''_t$ and a last linear layer with weight $w$. Let $f^{\mathrm{atlas-ofu}} = f(\cdot;\theta^{\mathrm{feat}}_{f''_t}, w_t)$, where $w_t$ is the weight maintained at round t by Algorithm 5. If we run Algorithm 1 with ATLAS in [7] and set up the step size pool $\mathcal{H} = \{\eta_i = O\left(\frac{\sigma}{\sqrt{KT}}\right) \cdot 2^{i-1}|i \in [N]\}$ ($N = 1 + \lceil \frac{1}{2}\log_2(1+2T)\rceil$), then under the same assumptions as Lemma 1 in [7], ATLAS-OFU satisfies that*

$$\mathbb{E}\left[\frac{1}{T}\sum_{t=1}^T \ell(f^{\mathrm{atlas-ofu}}; \mathcal{P}^{\mathrm{test}}_t) - \frac{1}{T}\sum_{t=1}^T \min_{w\in\mathcal{W}} \ell(f(\cdot;\theta^{\mathrm{feat}}_{f''_{t+1}}, w); \mathcal{P}^{\mathrm{test}}_t)\right] \leq O\left(\left(\frac{K^{1/3}}{\sigma^{2/3}}+1\right)\frac{V^{1/3}_{T,\ell}}{T^{1/3}} + \sqrt{\frac{K}{\sigma^2 T}}\right),$$
$$\tag{22}$$

*where $V_{T,\ell} := \sum_{t=2}^T \sup_{w\in\mathcal{W}} |\ell(f(\cdot;\theta^{\mathrm{feat}}_{f''_t}, w); \mathcal{P}^{\mathrm{test}}_t) - \ell(f(\cdot;\theta^{\mathrm{feat}}_{f''_{t-1}}, w); \mathcal{P}^{\mathrm{test}}_{t-1})|$, $\sigma$ denotes the minimum singular value of the invertible confusion matrices $\{C_{f''_1,D'_0}, \cdots C_{f''_T,D'_0}\}$ and $K$ is the number of classes and the expectation is taken with respect to randomness in the revealed co-variates.*

The proof is similar to that of Theorem 5, which mainly follows the similar idea in the original online ensembling framework [54] and ATLAS [7], and hence omitted.

**Discussion about the assumption.** In the theorems for UOGD and ATLAS, the definition of $V_{T,\ell}$ is shift severity from $\mathcal{P}^{\mathrm{test}}_t$. However, in the theorems for UOGD-OFU and ATLAS-OFU above, $V_{T,\ell}$ is shift severity from both $\mathcal{P}^{\mathrm{test}}_t$ and $\theta^{\mathrm{feat}}_{f''_t}$, which can be much larger. This might lead to harder convergence of the regret.

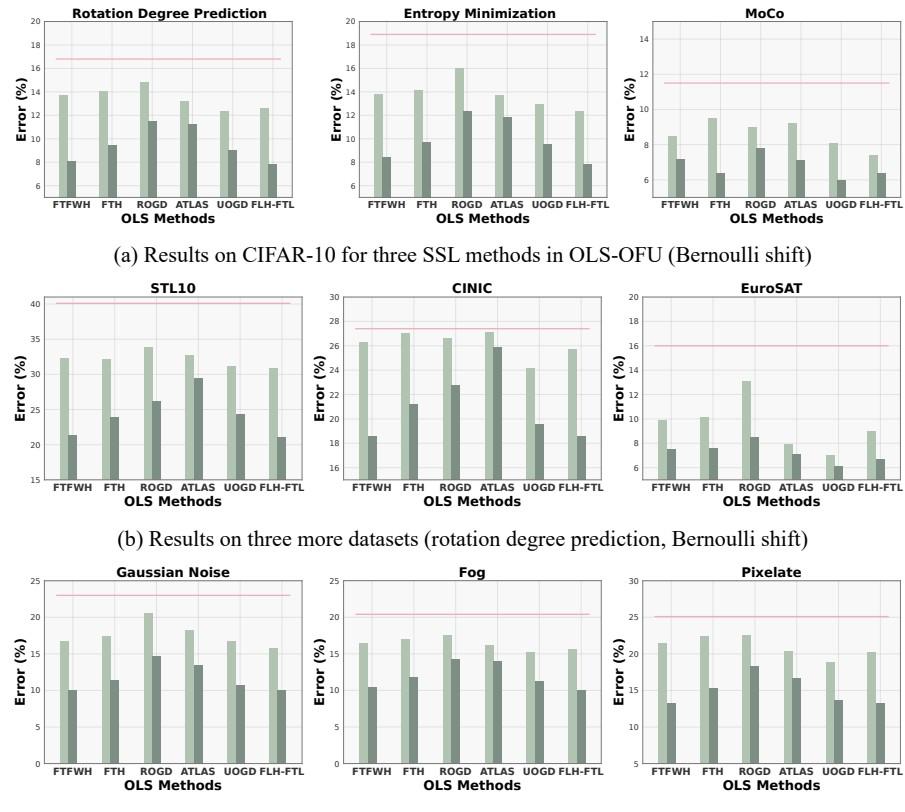

(a) Results on CIFAR-10 for three SSL methods in OLS-OFU (Bernoulli shift)

(b) Results on three more datasets (rotation degree prediction, Bernoulli shift)

(c) Results on three types of corruptions in CIFAR-10C (rotation degree prediction, Bernoulli shift)

Figure 4: Evaluation of OLS and OLS-OFU.

# E  Additional experiments

## E.1  Additional Details of Datasets, Online Shift Patterns, and SSL

**Severity of CIFAR-10C in the experiment.**  For each type of corruption in CIFAR-10C, we select a mild level of severity in the experiment section. Here we introduce the exact parameters of mild and high levels of severity for those corruptions. For Gaussian Noise, the severity level is 0.03. For Fog, the severity level is (0.75, 2.5). For Pixelate, the severity level is 0.75.

**Details of online shift patterns.**  Specifically, given two label distribution vectors $q$ and $q'$, we simulate the label marginal distributions at time $t$ as a weighted combination of them: $q_t := \alpha_t q + (1 - \alpha_t)q'$. In Sinusoidal shift, $\alpha_t = \sin \frac{i\pi}{L}$ (periodic length $L = \sqrt{T}$, $i = t \bmod L$) while in Bernoulli shift, $\alpha_t$ is a random bit (either 0 or 1), where the bit switches $\alpha_t = 1 - \alpha_{t-1}$ with probability $p = 1 - \frac{1}{\sqrt{T}}$. In our experiments, we set the initial distribution vector $q$ and $q'$ with $\frac{1}{K}(1, \cdots, 1)$ and $(1, 0, \cdots, 0)$. To sample the batch test data at time $t$, we first sample a batch of labels (not revealed to the learner) according to $q_t$. Then given each label we can sample an image from the test set, and collect this batch of images without labels as $S_t$.

**Details of self supervised learning techniques.**  *rotation degree prediction* involves initially rotating a given image by a specific degree from the set $\{0, 90, 180, 270\}$ and the classifier is required to determine the degree by which the image has been rotated. *Entropy minimization* utilizes a minimum entropy regularizer, with the motivation that unlabeled examples are mostly beneficial when classes have a small overlap. *MoCo* is a more advanced representation learning technique, using a query and momentum encoder to learn representations from unlabeled data by maximizing the similarity of positive pairs and minimizing the similarity of negative pairs.

| | FTFWH | FTH | ROGD | ATLAS | UOGD | FLH-FTL |
|---|---|---|---|---|---|---|
| OLS | 11.52% / 0.397 | 11.56% / 0.395 | 14.21% / 1.15 | 12.02% / 42.5 | 11.65% / 312 | 11.63% / 1.09 |
| OLS-OFU-BA ($\tau = 1$) | 10.5% / 260 | 10.5% / 264 | 14.% / 380 | 12.1% / 44.2 | 11.6% / 321 | 10.6% / 287 |
| OLS-OFU-BA ($\tau = 10$) | 9.19% / 28.1 | 9.64% / 28.7 | 13.2% / 40.9 | 12% / 43.9 | 10.6% / 319 | 9.39% / 28.4 |
| OLS-OFU-BA ($\tau = 50$) | 7.54% / 6.16 | 8.41% / 6.3 | 12.1% / 9.26 | 11.7% / 43.8 | 9.4% / 317 | 7.82% / 6.88 |
| OLS-OFU-BA ($\tau = 100$) | 6.79% / 3.4 | 7.57% / 3.38 | 10.9% / 5.21 | 10.5% / 43.6 | 8.46% / 314 | 7.04% / 4.09 |
| OLS-OFU-BA ($\tau = 500$) | 11.2% / 1.19 | 11.3% / 1.18 | 14.1% / 2.22 | 11.9% / 42.7 | 11.4% / 314 | 11.3% / 1.88 |

Table 4: Average error / time (minutes) of 6 original OLS methods versus OLS-OFU with various frequency $\tau$ in *batch accumulation*. The SSL in OLS-OFU is Entropy Minimization.

Their loss functions are as follow. When the SSL loss is rotation degree prediction, it requires another network $f^{\text{deg}}$ to predict the rotation degree, sharing the same feature extractor $\theta^{\text{feat}}$ as $f_0$ but with a different set of downstream layers. Its SSL loss $\ell_{\text{ssl}}(S; f)$ is defined as $\sum_{x \in S} \ell_{ce}(f^{\text{deg}}(R(x,i)), i)$, where $i$ is an integer uniformly sampled from $[4]$, and $R(x,i)$ is to rotate $x$ with degree $\text{DL}[i]$ from a list of degrees $\text{DL} = [0, 90, 180, 270]$. Alternatively, if the SSL loss is entropy minimization, $\ell_{\text{ssl}}(S; f)$ would be the entropy $\sum_{x \in S} \sum_{k=1}^{K} f(x)_k \log f(x)_k$. Moreover, the SSL loss of MoCo would be a contrastive loss (InfoNCE) where the positive example $x'$ is an augmented version of $x$ and other samples in the same time step can be the negative examples.

### E.2 Additional Results of OLS-OFU and Baselines

Figure 2 has three subfigures, showing the results of OLS-OFU with three SSL on CIFAR-10, the results on three more datasets for evaluating online label shift, and the results on CIFAR-10C for evaluating online generalized label shift; the online shift patterns are Sinusoidal shift for all these figures in Figure 2. We show similar figures in Figure 4 but now the online shift patterns are Bernoulli shift. We can have the exact same observations from Figure 4: the improvements from original OLS methods to OLS-OFU are significant and consistent across 6 OLS methods, 3 SSL techniques, 4 datasets with online label shift and 3 datasets with online generalized label shift.

We also evaluate three SSL methods in OLS-OFU on CIFAR-10C with mild severity for two online shift patterns. In Figure 7, the improvement from OLS to OLS-OFU is very significant but OLS-OFU cannot outperform OFU.

### E.3 Ablation Study on Frequency $\tau$ in OLS-OFU for Entropy Minimization and MoCo

Table 1 in the main paper shows the ablation study on $\tau$ when SSL is rotation degree prediction. Here we include Table 4 and Table 5 for the same ablation study on $\tau$ but the SSL in OLF-OFU is Entropy Minimization and MoCoin two tables respectively. We have same observations as what we observed from Table 1: 1. OLS-OFU with all $\tau$ outperforms OLS; 2. The average error of OLS-OFU decreases from $\tau = 1$ (*w/o* batch accumulation) to $\tau = 100$ and starts to increase after and this is because the effectiveness of SSL benefits from $\tau > 1$ while larger $\tau$ means less updates and hence hurts the long term performance; 3. the additional time cost of OLS-OFU with larger $\tau$ is smaller and $\tau = 100$ gives a great balance of performance and time cost.

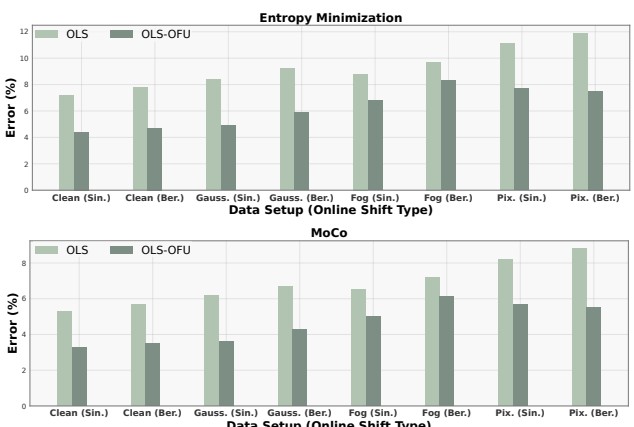

Figure 5: Empirical examination for the holdness of Equation 7. Clean denotes the experiment on CIFAR-10 and others denotes the corruption type. They are paired with two online shift patterns: Sinusoidal and Bernoulli.

|  | FTFWH | FTH | ROGD | ATLAS | UOGD | FLH-FTL |
|---|---|---|---|---|---|---|
| OLS | 6.21% / 0.392 | 7.03% / 1.05 | 8.23% / 0.888 | 7.82% / 42.2 | 6.97% / 302 | 6.45% / 0.44 |
| OLS-OFU-BA ($\tau = 1$) | 6.76% / 277 | 8.13% / 280 | 8.86% / 374 | 7.57% / 56.3 | 7.84% / 319 | 8.55% / 271 |
| OLS-OFU-BA ($\tau = 10$) | 5.64% / 28.6 | 7.08% / 29.7 | 8.01% / 39.7 | 7.2% / 44.7 | 6.87% / 313 | 7.2% / 28.5 |
| OLS-OFU-BA ($\tau = 50$) | 4.52% / 6.18 | 6.03% / 6.96 | 7.16% / 8.77 | 6.84% / 43.7 | 5.9% / 312 | 5.86% / 6.09 |
| OLS-OFU-BA ($\tau = 100$) | 3.97% / 3.42 | 5.3% / 4.2 | 6.3% / 4.91 | 6.01% / 43.4 | 5.19% / 308 | 5.15% / 3.36 |
| OLS-OFU-BA ($\tau = 500$) | 5.06% / 2.13 | 6.32% / 2.86 | 7.52% / 3.2 | 7.12% / 43.1 | 6.23% / 309 | 6.05% / 2.12 |

Table 5: Average error / time (minutes) of 6 original OLS methods versus OLS-OFU with various frequency $\tau$ in *batch accumulation*. The SSL in OLS-OFU is MoCo.

### E.4 Empirical Validation of Equation 7

Similar to Figure 3, as shown in Figure 5, it is evident that OLS-OFU with SSL chosen as Entropy Minimization and MoCo is able to yield improvements on the *baseline* of the regret as shown in Equation 7.

### E.5 Self-training

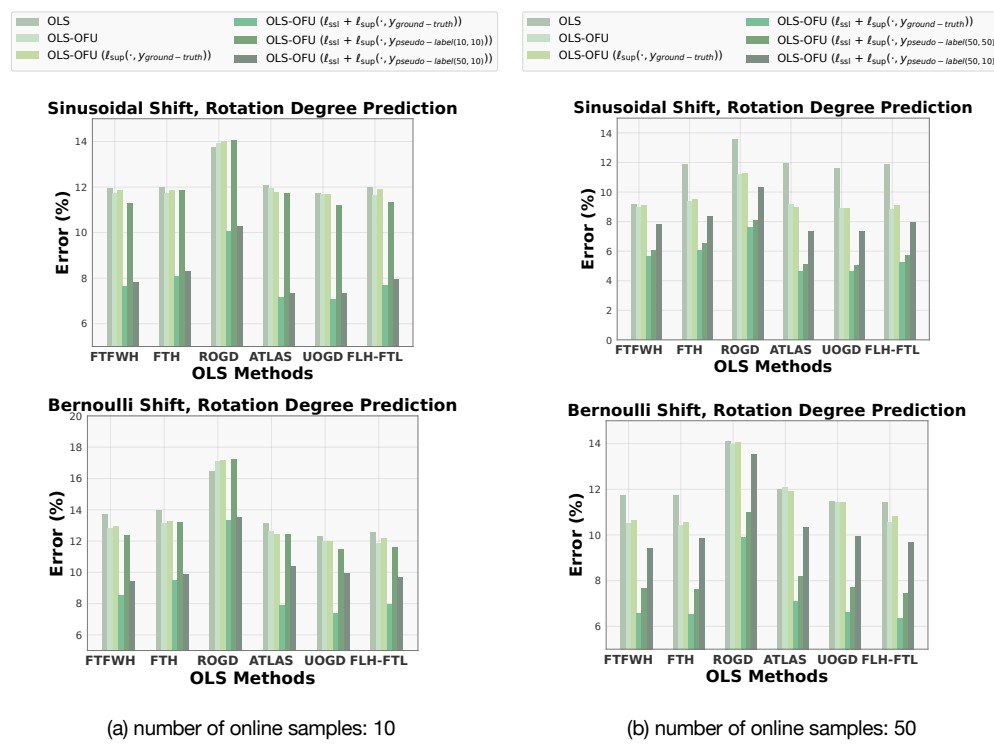

(a) number of online samples: 10

(b) number of online samples: 50

Figure 6: Results on pseudo-labelling.

Pseudo labelling [31], a common self-training technique, generates pseudo labels for unlabelled data and uses them to update the model. Though we are not able to use ground-truth labels to compute feature extractor updates, we can use the model at time $t$ to make predictions with respect to the online samples at time $t$, and train on the inputs with their assigned (pseudo) labels. An issue that arises in self-training is confirmation bias, where the model repeatedly overfits to incorrect pseudo-labels. As such, different methods can be used to select which samples will be pseudo-labelled and used in updating the model, e.g. using data augmentation [3], using regularization to induce confident low-entropy pseudo-labelling [17], using softmax thresholds to filter out noisy low-confidence predictions [50]. We make use of ensembles to identify noisy low-confidence/entropy pseudo-label predictions,

though other various alternatives can also be used. In addition to OLS and OLS-OFU, we highlight the methods under comparison:

- *OLS-OFU ($\ell_{\mathrm{sup}}(\cdot, y_{\mathrm{ground\text{-}truth}})$)*: Instead of computing pseudo-labels, we make use of the correct ground-truth labels $y_{\mathrm{ground\text{-}truth}}$. Recall $\ell_{\mathrm{sup}}$ is the supervised learning loss. We update the feature extractor with the supervised loss w.r.t. ground-truth labels $\ell_{\mathrm{sup}}(\cdot, y_{\mathrm{ground\text{-}truth}})$.

- *OLS-OFU ($\ell_{\mathrm{ssl}} + \ell_{\mathrm{sup}}(\cdot, y_{\mathrm{ground\text{-}truth}})$)*: Instead of computing pseudo-labels, we make use of the correct ground-truth labels $y_{\mathrm{ground\text{-}truth}}$. Recall $\ell_{\mathrm{ssl}}$ and $\ell_{\mathrm{sup}}$ are the self-supervised and supervised learning losses respectively. We update the feature extractor with both the self-supervised loss $\ell_{\mathrm{ssl}}$ as well as the supervised loss w.r.t. ground-truth labels $\ell_{\mathrm{sup}}(\cdot, y_{\mathrm{ground\text{-}truth}})$.

- *OLS-OFU ($\ell_{\mathrm{ssl}} + \ell_{\mathrm{sup}}(\cdot, y_{\mathrm{pseudo\text{-}label(\#samples=, \#FU\text{-}samples=)}})$)*: Recall $\ell_{\mathrm{ssl}}$ and $\ell_{\mathrm{sup}}$ are the self-supervised and supervised learning losses respectively. We compute pseudo-labels $y_{\mathrm{pseudo\text{-}label}}$, and update the feature extractor with both the self-supervised loss $\ell_{\mathrm{ssl}}$ as well as the supervised loss w.r.t. pseudo-labels $\ell_{\mathrm{sup}}(\cdot, y_{\mathrm{pseudo\text{-}label}})$.

**How to compute pseudo-labels?**    We now describe the procedure to compute pseudo-labels for $\ell_{\mathrm{sup}}(\cdot, y_{\mathrm{pseudo\text{-}label(\#samples=, \#FU\text{-}samples=)}})$. The seed used to train our model is 4242, and we train an additional 4 models on seeds 4343, 4545, 4646, 4747. With this ensemble of 5 models, we keep sampling inputs at each online time step until we have `#FU-samples` samples, or we reach a limit of `#samples` samples. We accept an input when the agreement between the ensembles exceeds a threshold $e = 1.0$ (i.e. we only accept samples where all 5 ensembles agree on the label of the online sample). In the default online learning setting, there are only `#samples=10`, and therefore there may not be enough accepted samples to perform feature update with, thus we evaluate with a continuous sampling setup, where we sample `#samples=50` (and evaluate on all these samples), but only use the first 10 samples (`#FU-samples=10`) to perform the feature extractor update.

**Results on pseudo-labelling.**    We report the results of OLS-OFU with $\tau = 1$ in this section. First, we find that *OLS-OFU ($\ell_{\mathrm{ssl}} + \ell_{\mathrm{sup}}(\cdot, y_{\mathrm{ground\text{-}truth}})$)* attains the lowest error and is the lower bound we are attaining towards. Evaluating *OLS-OFU ($\ell_{\mathrm{ssl}} + \ell_{\mathrm{sup}}(\cdot, y_{\mathrm{pseudo\text{-}label(\#samples=10, \#FU\text{-}samples=10)}})$)*, we find that the performance does not outperform OLS-OFU, and is not near *OLS-OFU ($\ell_{\mathrm{ssl}} + \ell_{\mathrm{sup}}(\cdot, y_{\mathrm{ground\text{-}truth}})$)*. If we set the threshold $e$ too high, there may not be enough online samples to update the feature extractor. If we set the threshold $e$ too low, there may be too many incorrect labels and we incorrectly update our feature extractor. As such, we would like to sample more inputs at each online time step such that we can balance this tradeoff. We sample `#samples=50` at each online time step, and update with `#FU-samples` $\leq$ `10`. For fair comparison, we also show the comparable methods in both `#samples=10, #FU-samples=10` and `#samples=50, #FU-samples=50` settings.

With this sampling setup, we find that *OLS-OFU ($\ell_{\mathrm{ssl}} + \ell_{\mathrm{sup}}(\cdot, y_{\mathrm{pseudo\text{-}label(\#samples=50, \#FU\text{-}samples=10)}})$)* can outperform both *OLS-OFU (`#samples=10`)* and *OLS-OFU (`#samples=50`)*. Though it does not exceed neither *OLS-OFU ($\ell_{\mathrm{ssl}} + \ell_{\mathrm{sup}}(\cdot, y_{\mathrm{ground\text{-}truth}})$)* for `#samples=10` nor `#samples=50`, it lowers the gap considerably.

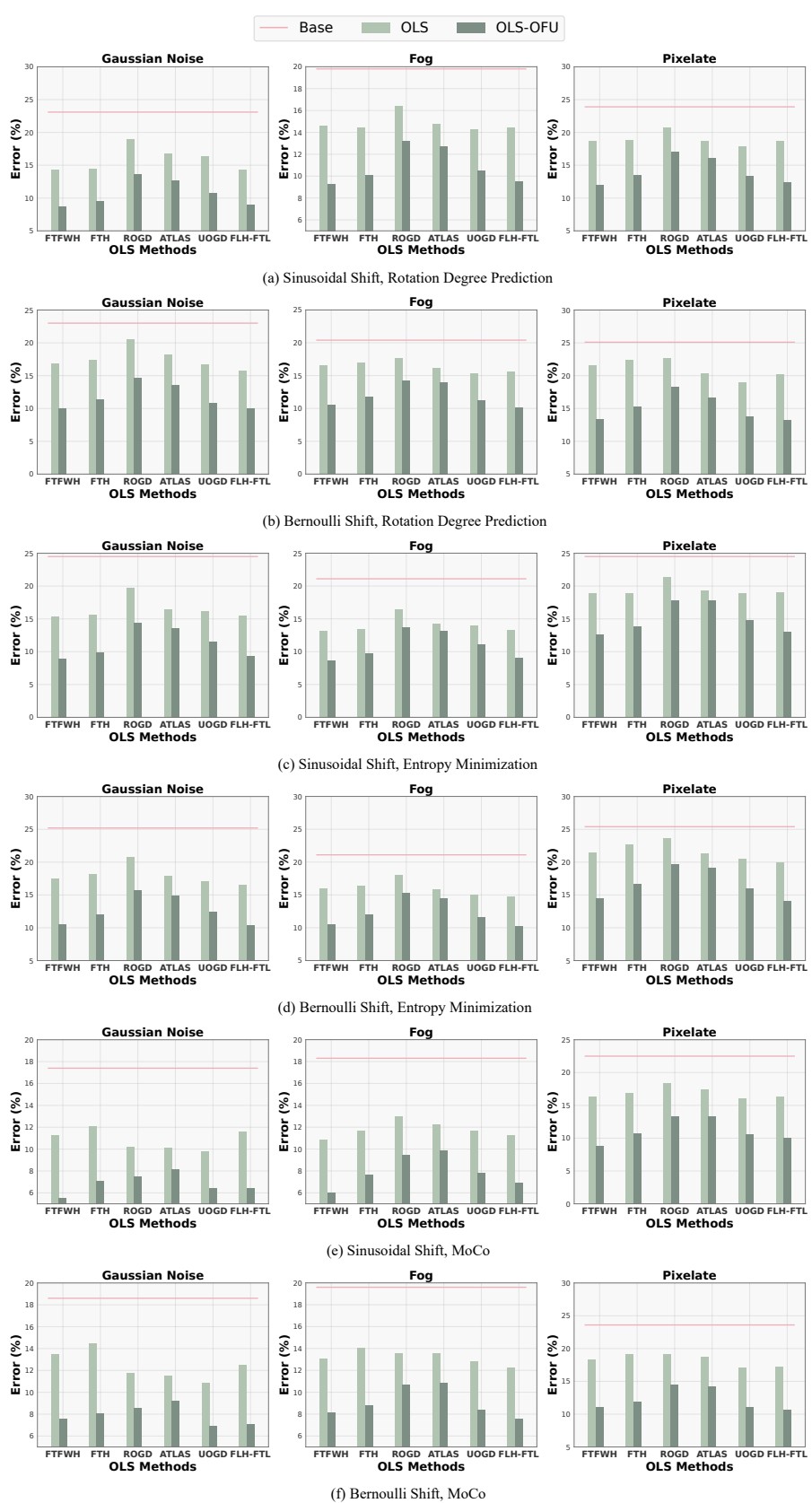

Figure 7: Results of two online shift patterns on CIFAR-10C and three SSL methods in OLS-OFU.

