# OpenReview forum: "Online Feature Updates Improve Online (Generalized) Label Shift Adaptation"
_NeurIPS.cc/2024/Conference — NeurIPS 2024 poster_

### Official Review · Reviewer_Dj3x · 2024-07-11

**Soundness:** 3
**Presentation:** 2
**Contribution:** 2
**Rating:** 5
**Confidence:** 4

**Summary:**

The authors of this paper focus on the task of label distribution shift in the online setting without true labels. Following the importance of improving feature extractors even during test-time found in the current literature, they propose to update the feature extractor online with unlabeled instances during testing for online label shift and then refine the last classification layer with labeled instances. The proposed method is simple and intuitive, and extensive experiments over several label shift cases and benchmark datasets have shown its effectiveness.

**Strengths:**

1.	The online label shift task is an interesting problem and worth paying more attention to, and the proposed method in this paper is simple and intuitive.
2.	In addition, this paper has proved the regret convergence of the proposed method partially, so that provides the necessary justification.
3.	The experimental section and the supplementary material present experimental results that demonstrate the effectiveness of the method proposed in the article.

**Weaknesses:**

1.	The motivation of this paper is not clear. The authors are just motivated by “the potential for improving feature extractors”, and hypothesize “that a similar effect can be harnessed in the context of (generalized) label shift”. This may be too intuitive. Why is improving feature extractors helpful for only label shift? What can improve feature extractors gain? Are there some theoretical or empirical results to prove that? The theory of regret convergence may be not enough.
2.	The novelty of the proposed method may be limited. Improving feature extractors has been proved its importance during testing time for distribution shifts in the current literature, such as [1]. What is the difference between the proposed method and them? What are your main contributions in this paper?
3.	What are the limitations and broader impacts?

[1] Y. Sun, X. Wang, Z. Liu, J. Miller, A. Efros, and M. Hardt. Test-time training with self-supervision for generalization under distribution shifts. In International conference on machine learning, pages 9229–9248. PMLR, 2020.

**Questions:**

1.	What is the difference between (online) label distribution shift and domain adaption? Can the proposed method adapt to the domain adaption task?
2.	What will happen if there are some instances related to new unseen categories during testing? And how to handle this case? Can improving the feature extractor be helpful?

**Limitations:**

See the above “*Weaknesses*” and “*Questions*” parts.

---

> ### Author Rebuttal · Authors · 2024-08-07
>
> We thank the reviewer for providing insightful comments! Here are our responses.
>
> **Motivation and its justification**. We would like to answer your questions in this point separately.
> - “Why is improving feature extractors helpful for only label shift?” In fact, updating feature extractor can help with other types of shift too. As discussed at line 359-361, the existing algorithm to solve online *covariate* shift, which has strong theoretical guarantee, also doesn’t take feature extractor update into account. We believe by adopting online feature update, the performance of the algorithm can get certain improvement as well, and our work shows this possibility as a first work to introduce feature extractor update into those theoretical algorithms.
> - “What can prove feature extractors gain? Are there some theoretical or empirical results to prove that?”  Equation (7) describes **the gain from the feature extractor improvement rather than the label shift adaptation**, because it is “almost” the best loss at time $t$ given the updated feature extractor $f_t’’$ or the old feature extractor $f_0$ together with ** the knowledge of label distribution $q_t$**. On the one hand, the theoretical guarantee can be improved from the feature extractor when Equation (7) holds; on the other hand, in the experiment, we empirically validate the holdness of equation 7. We believe these two theoretical and empirical evidence explain how the feature extractor updates improve our target problem.
>
> We will write those arguments more explicitly in our revision!
>
> **Relation to the related work [1]**. Thanks for connecting our work to [1]. First, as cited in section 2, we introduce it as a self-supervised feature update algorithm, which can serve for our motivations discussed at line 135-141. Moreover, we discussed in Section 3.4 on how it is particularly powerful to address online *generalized* label shift. We would like to clarify this point here:
>
> According to the definition of generalized label shift, it can be reduced to label shift once the learner knows the underlying feature mapping $h$, and online feature update actually helps learn the underlying feature mapping $h$ that makes $P(h(x_t)|y_t)$ invariant. In the literature of test-time training [1] and more [2, 3], they found the self-supervised learning of feature updates can explicitly align the feature space of new distribution to the feature space of training distribution; they even only trained with the feature extractor **without retrain the last linear layer**, this strongly supports the straightforward feature space alignment between original distribution and shifted distribution.
>
> We would like to add this explanation in Section 3.4, which we believe can better explain the role of [1] when tackling the online *generalized* label shift.
>
> **Our main contribution**. Our main contribution is as a first work to bring self-supervised learning into the online label shift problem, which was mainly studied from the theoretical aspects. Our algorithm provides a way to leverage self-supervised learning, which largely improves the performance as validated through the experiments and still keeps the similar theoretical guarantees.
>
> **Limitations and broader impacts**. The goal of this work is to advance the robustness of models for real-world deployment. Our work contributes to the mitigation of different adverse effects of online label shift, such as out-of-date or miscalibrated models (e.g. healthcare or finance settings, autonomous systems such as self-driving). Adaptation to changing shifts has positive ethical implications, such as in fairness (e.g. improving the models with updated, fair data over time) or privacy (e.g. unlearning data owned by specific groups).
>
> **Difference between label distribution shift and domain adaptation**. Domain adaptation is a broader concept, including the case where the support of x can be totally different. Label distribution shift has an explicit assumption where $P(x|y)$ doesn’t change. Our algorithm and previous label shift adaptation algorithms are studied given this explicit assumption and the theoretical results are based on this assumption. While those algorithms work well for label shift, there is no guarantee that they can work well without the label shift assumption.
>
> **How to handle unseen categories**. This is actually an interesting question! As pointed out by Reviewer aV82, [4] focuses on this special setting by unsupervised estimating the portion of unseen data. We will add the discussion of this setting and the related work [4] into the related work section.
>
> [1] Y. Sun, X. Wang, Z. Liu, J. Miller, A. Efros, and M. Hardt. Test-time training with self-supervision for generalization under distribution shifts. In International conference on machine learning, pages 9229–9248. PMLR, 2020.
>
> [2] Wang, Dequan, et al. "Tent: Fully test-time adaptation by entropy minimization." arXiv preprint arXiv:2006.10726 (2020).
>
> [3] Liu, Yuejiang, et al. "Ttt++: When does self-supervised test-time training fail or thrive?." Advances in Neural Information Processing Systems 34 (2021): 21808-21820.
>
> [4] Qian Y Y, Bai Y, Zhang Z Y, et al. Handling New Class in Online Label Shift[C]//2023 IEEE International Conference on Data Mining (ICDM). IEEE, 2023: 1283-1288.

---

> > ### Comment · Reviewer_Dj3x · 2024-08-11
> >
> > Thanks for your responses. I have read them, and would like to keep my score.

---

### Official Review · Reviewer_aV82 · 2024-07-12

**Soundness:** 3
**Presentation:** 3
**Contribution:** 3
**Rating:** 6
**Confidence:** 4

**Summary:**

This paper addresses the problem of online label shift and proposes a novel algorithm that exploits feature representation learning to enhance performance. Inspired by test-time training literature, the proposed method uses self-supervised learning to refine the feature extraction process. The algorithm comes with strong theoretical guarantees, and experiments demonstrate its effectiveness.

**Strengths:**

Online label shift problem is a common but crucial issue in many real-world applications. This paper considers the important task of exploiting feature representation, proposing a novel algorithm that leverages self-supervised learning to refine the feature extraction process in the online label shift problem.

The proposed method satisfies the practical requirements while having solid theoretical guarantees that ensure its general applicability and reliability in various non-stationary learning scenarios.

Experiments show the superiority of the proposed algorithm on several benchmark datasets and two distinct online shift patterns, highlighting its effectiveness and practical impact.

**Weaknesses:**

1. The proposed method requires the storage of previous historical data for self-supervised learning (batch accumulation in the paper). This may not be feasible in certain problems with privacy concerns.

2. The experiments are primarily conducted on simulated benchmark datasets, such as CIFAR-10 and CINIC, rather than real-world applications or datasets. This limits the understanding of the method's performance in practical settings.

3. It is recommended to include some recent work on online label shift in the paper, such as [1], which addresses the appearance of new class data in the scenario of online label shift.


[1] Qian Y Y, Bai Y, Zhang Z Y, et al. Handling New Class in Online Label Shift[C]//2023 IEEE International Conference on Data Mining (ICDM). IEEE, 2023: 1283-1288.

**Questions:**

See weakness above.

In addtion, for the storage of historical data, is it feasible to apply data augmentation techniques to the limited number of data per time stamp and use the augmented data for self-supervised learning? How does the number of historical data stored affect the proposed algorithm?

**Limitations:**

the authors adequately addressed the limitations

---

> ### Author Rebuttal · Authors · 2024-08-07
>
> We thank the reviewer for providing insightful comments! Here are our responses.
>
> **The storage of previous historical data**. This is actually an insightful point! We acknowledge this can bring additional privacy concerns, depending on the practical scenarios, and we would like to discuss this in the revision. At the same time, as shown in Table 1, we can find that even without batch accumulation (batch size $\tau$=1), OLS-OFU still outperforms OLS. Parameter $\tau$ can be treated as the performance and privacy trade-off.
>
> **Dataset set-up**. Our selection of datasets and the way to simulate the shift pattern mainly follow the previous online label shift literature [1, 2] and additionally, we experiment with additional domain adaptation dataset CIFAR-10C. Although we agree experimenting with real-world shifts can be meaningful, we believe our current experiments are sufficient enough to prove the improvement from the literature.
>
> **Related work**. Thank you for the pointer! This work particularly solves the unseen categories in the online label shift setting and is very relevant. It tackles the new unseen class by unsupervised estimating the portion of unseen data. We will add this literature in our related work section.
>
> [1] Baby, Dheeraj, et al. "Online label shift: Optimal dynamic regret meets practical algorithms." Advances in Neural Information Processing Systems 36 (2024).
>
> [2] Bai, Yong, et al. "Adapting to online label shift with provable guarantees." Advances in Neural Information Processing Systems 35 (2022): 29960-29974.

---

> > ### Comment · Reviewer_aV82 · 2024-08-12
> >
> > Thank you for your reply. After reading other comments and rebuttals, I would like to keep my score.

---

### Official Review · Reviewer_XJxy · 2024-07-14

**Soundness:** 2
**Presentation:** 1
**Contribution:** 2
**Rating:** 3
**Confidence:** 4

**Summary:**

This paper addresses the online label shift (OLS) adaptation problem, which involves continually adapting a pretrained offline model to test data with various and evolving label distribution shifts. The proposed method integrates existing self-supervised learning (SSL) techniques into current OLS methods, based on three proposed method-combination principles. Experimental results demonstrate that incorporating SSL methods leads to performance improvements.

**Strengths:**

1. The proposed method encompasses various OLS baseline methods and self-supervised learning approaches, demonstrating a comprehensive range of experimental cases.

2. The proposed method consistently achieves performance improvements over the comparison methods.

**Weaknesses:**

1. **Limited Novelty and Significance**: The proposed method appears to be a straightforward combination of existing approaches, specifically SSL and OLS methods. The three proposed principles for combining these approaches are relatively trivial:

    - Since SSL affects extracted features, OLS is performed first.

    - As SSL changes the feature extractor, the classifier is retrained.

    - Given that SSL requires additional training resources, it is applied after accumulating enough data.

    These principles seem too basic to be considered significant technical contributions.

2. **Problems in Paper Structure**: The paper has serious structural issues.
    The proposed method relies heavily on existing OLS methods, yet there is a lack of brief or detailed introduction of related OLS methods. Only three sentences describe OLS and SSL methods.

    Conversely, the problem setting of online label shift, which can be summarized in one sentence as "the test data have a different label distribution from training data", is explained with excessive and redundant content, including irrelevant details about online distribution shift.

    As a result, the experimental section is compressed into two pages, limiting the space to present results adequately. The conclusion is also excessively brief, reduced to just one sentence.

3. **Unreasonable Problem Setting**: The addressed problem seems unreasonable for several reasons.

    - **Data Privacy Concern**: Adapting the model at each time step requires storing all historical training data, testing data, and models, raising significant data privacy concerns. For instance, in the MRI example mentioned in the paper, it is questionable whether you are allowed to carry MRI data from 999 clinics just to adapt the model to the 1000th clinic.

    - **Adequate Offline Training Data**: The offline training data appears overly sufficient, making the problem less challenging. In the CIFAR-10 experiment, all training data are used for offline training, with the only challenge being the variation in label distribution in the test data. Without any adaptation, the model performs well on the testing data.

4. **Low Baseline Performance**: The reported baseline result without any adaptation appears too low. A ResNet-18 model on CIFAR-10 typically achieves around 93\% accuracy on test data without any special data augmentation (see https://github.com/kuangliu/pytorch-cifar), which is significantly higher than the reported 84\% and most adaptation results.

5. **Minor Typos**:
    - Line 83: How do you reweight a model f? Do you mean reweight the model output?
    - Line 179: OGD or ROGD?
    - Lines 232-233: "Either...or" should be used correctly.
    - Line 249: Should it be "is" or "as"?

**Questions:**

See Weaknesses.

**Limitations:**

Already addressed by Authors.

---

> ### Author Rebuttal · Authors · 2024-08-07
>
> We would like to thank the reviewer for providing some useful comments about data privacy concerns and some typos. However, we respectfully disagree with the criticism on novelty, paper structure and the experiment set-up (data split and baseline performance). We reply to the weaknesses and questions point by point as below.
>
> **Novelty of three principles**. While we agree that it turns out the steps are simple, the choice of them rather than other options is out of careful considerations. We list the possible other options and re-iterate why our final design stands out from other options.
> 1. *Update the feature extractor first or run original OLS first*. Through our analysis, we found only running original OLS first can still guarantee the theoretical results, while the other option cannot.
> 2. *Re-training the linear layer under training distribution or other distribution*. A more natural option is actually to retrain the linear layer with the most recent estimated label distribution. However, by checking with the original OLS methods, they require the model to be in the training distribution, or at least a distribution having all classes, but test distribution is not guaranteed to have this property since it can be just one class.
> 3. *Batch accumulation or not*. In fact, as shown in Table 1, even with batch size $\tau=1$ OLS-OFU has consistent improvement over OLS, and the necessity of batch accumulation can be missed. Through the experiment, as shown in Table 1, the benefit of batch accumulation brings both the performance gain and more time efficiency.
> Moreover, through theoretical and empirical analysis, we show that our algorithm provides a way to leverage self-supervised learning, which largely improves the performance as validated through the experiments and still keeps the similar theoretical guarantees.
>
> **Paper structure**.
> > “Only three sentences describe OLS and SSL methods.”
>
> We actually spent more energy to describe OLS and SSL methods. Given that our focus is on how our algorithm makes the bridge between the two, we included the sufficient details in the main paper and enumerated more details in the appendix.
> - For OLS, we kindly refer the reviewer to these positions:
>     1. At line 115 - 125 in Section 2, original OLS methods do not update feature extractor and this motivates the methodology of our paper.
>     2. Line 175-181 describes the details of the condition where the theoretical guarantees would hold for original OLS methods, and this motivated our first design of our algorithm.
>     3. Line 187-191 describes the choice of hypothesis space of original OLS methods, and the underlying reason for this choice in the literature motivates our second design of our algorithm.
>     4. Moreover, for completeness, we included detailed algorithms of all 5 OLS methods as Algorithm 2-5 in the appendix.
> - For SSL, the details of SSL are not our focus,  as long as it has the form of $\ell_{\rm ssl}(S)$ for any batch of data $S$. Our algorithm OLS-OFU should work for general SSL methods, which is one of the information we would like to convey from our experiments. Therefore, we experiment with different SSLs as introduced at line 304-308 and their numerous details in the Appendix E.1.
>
> > “The problem setting of online label shift, which can be summarized in one sentence as "the test data have a different label distribution from training data", is explained with excessive and redundant content, including irrelevant details about online distribution shift.”
>
> We kindly disagree with the description of online distribution shift and online label shift is redundant. Rather than just describe the mathematical problem, other components are necessary in introducing a problem too to make the paper have a broader audience, which include the motivating example (line 85-96), the description of objective function in the online setting (Line 97-103), the learning setting of unsupervised adaptation and limited unlabeled batch (Line 104-110), the assumption of label shift (Line 110-113), the summarization of existing online label shift methods and their limitations (Line 114-128) and introducing online generalized label shift adaptation as a first work (line 128 until line 129.)
>
> > “The experimental section is compressed into two pages, limiting the space to present results adequately”
>
> We believe our experiment results are strong enough to demonstrate the effectiveness of our algorithms, which can be reflected by other reviews. We disagree that a two-pages experiment is a limitation, while we would appreciate any *concrete advice* to improve our result presentation.
>
> **Data privacy concern**. This is actually an insightful point. We acknowledge this can bring additional privacy concerns, depending on the practical scenarios, and we would like to discuss this in the revision. At the same time, as shown in Table 1, we can find that even without batch accumulation (batch size $\tau$=1), OLS-OFU still outperforms OLS. Parameter $\tau$ can be treated as the performance and privacy trade-off.
>
> **Data split and baseline performance**. There seems to be some misunderstanding about our data split and baselines, where we basically followed the literature. For the data split, as described at Line 293, our model was trained by 80% training set offline, with a 20% split as a validation set. We followed the code released from [1] for training the $f_0$ and our performance of “base” matches their numbers in Table 1, which are around 16% error for the CIFAR10.
>
> **Typos**. Thanks for catching them! We will fix the typos. The definition of reweight is to reweight the soft-max probability by element-wise multiplying another vector.
>
> [1] Baby, Dheeraj, et al. "Online label shift: Optimal dynamic regret meets practical algorithms." Advances in Neural Information Processing Systems 36 (2024).

---

> > ### Comment · Reviewer_XJxy · 2024-08-12
> >
> > Thanks for you response. Despite the feedback from the authors, most of the problems in this paper remain there.
> >
> > **Limited novelty and significance.** The proposed method is a simple combination of existing OLS and SSL methods. The main contribution comes from the proposed three combination principles. While we agree that other options are not optimal, these principles are too trivial and lack of significance.
> >
> > **Paper structure.**
> > - Counting all the sentences the author mentioned, there are totally 5 sentences introducing the related works about OLS methods.
> > - This paper has nothing to do with online distribution shift, including the detailed description about it instead of OLS methods is unreasonable and misleading, especially given that the proposed method heavily depends on previous OLS methods. We would not mind including online distribution shift if existing OLS methods have been thorougly described.
> > - One paragraph for the analysis of experimental results from line 317 to 330 are far away from enough. One sentence describing three figures are far away from enough.
> > - One sentence for conclusion is far away from enough.
> >
> > **Data privacy concern.** The data privacy concern is raised not only from the batch accumulation, but also the requirement of the offline training data, all the training data from previous time step (<t). Here comes the same question, are you allowed to carry MRI data from the first 999 clinics just to adapt the model to the 1000th clinic?
> >
> > **Too much offline training data.**
> > - Both training and validation sets count as data used for training. Thus, all the training data from CIFAR-10 has been used for offline model training, leaving the addressed OLS setting unchallenging at all.
> >
> > **Low baseline performance**
> > - The reported low baseline accuracy has not been explained. From the included link (https://github.com/kuangliu/pytorch-cifar), ResNet-18 on CIFAR-10 achieves 93\% accuracy. From the original ResNet paper, ResNet-18 on CIFAR-10 achieves an accuracy of 91.25\%. CIFAR-10 dataset has its own fixed testing set. Without any adaptation, the offline model outperforms almost all the reported results. This raises the question, is OLS adaptation necessary? Why wouldn't we train the offline model adequately in advance?
> >
> > Based on the forementioned reasons, I would recommend a clear Reject for this submission.

---

### Official Review · Reviewer_ZntR · 2024-07-31

**Soundness:** 3
**Presentation:** 3
**Contribution:** 2
**Rating:** 5
**Confidence:** 3

**Summary:**

This paper introduces a novel method for addressing label shifts in an online setting, where data distributions change over time, and obtaining timely labels is challenging. Unlike traditional approaches, this paper explores enhancing feature representations using unlabeled data during test time. The proposed method, Online Label Shift adaptation with Online Feature Updates (OLS-OFU), leverages self-supervised learning to refine the feature extraction process, thereby improving the prediction model. This approach is designed to maintain similar online regret convergence to existing results while incorporating improved features. Empirical evaluations show that OLS-OFU achieves substantial improvements over current methods, demonstrating that integrating online feature updates is as effective as the fundamental online label shift methods themselves. The results are consistent across various datasets and scenarios, highlighting the robustness and generality of OLS-OFU. The paper suggests that this method could be extended to more complex scenarios, such as online covariate shift and varying domain shifts over time.

**Strengths:**

-	In the Problem Setting & Related Work section, the paper effectively describes the mathematical definition of online label shift (OLS) adaptation and related research. By leveraging a similar mathematical framework, the paper clearly explains the approach to solving the online generalized label shift adaptation problem, demonstrating that the proposed method is well-motivated and thoroughly explained.
-	The author thoroughly explores potential questions and concerns associated with the introduction of new methods for addressing the problem. The paper presents well-defined principles and explanations that connect these concerns to the proposed approach, demonstrating a deep and thoughtful analysis of the new methodology
-	The experimental results presented in the paper effectively showcase the strong performance of the proposed OLS-OFU method compared to both baseline approaches and the existing OLS method. The comparisons clearly illustrate that OLS-OFU achieves better results, validating the efficacy of the proposed technique.

**Weaknesses:**

- From a critical perspective, the online generalized label shift problem presented in the paper could be viewed as a sub-field of concept drift, where the relationship between x and y changes over time. Therefore, a detailed explanation of the differences between the proposed online generalized label shift scenario and the traditional concept drift is required. Additionally, it would be beneficial to conduct experimental comparisons to determine whether methods developed for concept drift could be effective in the experimental settings used in this study.

**Questions:**

- What is the difference between concept drift and online generalized label shift situations?

---

> ### Author Rebuttal · Authors · 2024-08-07
>
> We thank the reviewer for providing insightful comments! Here are our responses.
>
> **Discussion with concept drift.** Yes, you are right that the (online) generalized label shift problem is a sub-field of concept drift and has its particular assumption. According to the definition of generalized label shift, it can be reduced to label shift once the learner knows the underlying feature mapping $h$. We would like to explain how each component of our method can solve each part specifically according to the literature and hence our method .
> 1. Online feature update learns the underlying feature mapping $h$ that makes $P(h(x_t)|y_t)$ invariant. In the literature of test-time training [1, 2, 3], they found the self-supervised learning of feature updates can explicitly align the feature space of new distribution to the feature space of training distribution; they even only trained with the feature extractor **without retrain the last linear layer**, this strongly supports the straightforward feature space alignment between original distribution and shifted distribution.
> 2. Built upon the updated feature extractor from 1, the problem can be reduced to online label shifts and we adopt the particular online label shift method, which follows the well-studied offline label shift method.
> Thank you for raising this point about generalized label shifts. We will add these explanations to Section 3.4 and we believe this can help improve the understanding of how our method can tackle the online generalized label shift.
>
> [1] Y. Sun, X. Wang, Z. Liu, J. Miller, A. Efros, and M. Hardt. Test-time training with self-supervision for generalization under distribution shifts. In International conference on machine learning, pages 9229–9248. PMLR, 2020.
>
> [2] Wang, Dequan, et al. "Tent: Fully test-time adaptation by entropy minimization." arXiv preprint arXiv:2006.10726 (2020).
>
> [3] Liu, Yuejiang, et al. "Ttt++: When does self-supervised test-time training fail or thrive?." Advances in Neural Information Processing Systems 34 (2021): 21808-21820.

---

> > ### Comment · Reviewer_ZntR · 2024-08-11
> >
> > Thanks for your responses. I have read them and would like to keep my score.
> >
> > Here are some clarifications for my score: I still have doubts regarding the experimental validity of concept drift. So, I leaned toward the accept side but stayed on the borderline.

---

### Official Review · Reviewer_fST3 · 2024-08-12

**Soundness:** 2
**Presentation:** 3
**Contribution:** 3
**Rating:** 7
**Confidence:** 3

**Summary:**

This paper addresses the problem of online (generalized) label shift, where label information is unavailable during testing, and the label distributions change over time. The main contribution of this work is the proposal of a unified framework that integrates feature learning into the online learning process, enabling the method to leverage the strengths of deep models. Theoretical analysis and experiments demonstrate the effectiveness of the proposed approach

**Strengths:**

Overall, this paper is well-motivated and makes a valuable contribution to the online label shift problem. It is an important question for me on how to effectively incorporate feature learning into the online learning process, and this paper provides a unified framework that demonstrates strong empirical performance. The strengths of this paper are as follows:

+ The paper presents a general method applicable to various online label shift approaches proposed in the literature.
+ The experimental results show a significant improvement in classification accuracy by incorporating the feature learning process.
+ The proposed method is robust to the generalized label shift problem.

**Weaknesses:**

- One of my main concerns is the theoretical analysis of the feature learning component. While Theorem 1 is commendable for demonstrating that the proposed method is comparable to the best model adapted from $f_t^{''}$, it remains unclear how effective the feature extractor obtained via self-supervised learning is (from a theoretical view). The self-supervised learning technique appears to be used as a black box, with no theoretical insight provided into its performance.

- Regarding storage costs: It seems that the proposed method requires storing the training set $D_0$. Such a requirement is somewhat unfavorable in practice, as the training set $D_0$ typically consists of a large volume of data.

- Concerning Principle 1: I understand that Principle 1 is essential for achieving theoretical guarantees in the online label shift problem. However, I am uncertain whether such a requirement is necessary in practice. Should the update procedure depend on the amount of data available at each round? If we have a reasonable amount of data at each time, wouldn't it be more reasonable to update the feature extractor before the online learning process to gather more information? Conversely, if the data at each iteration is limited, I'm unsure if the difference between update procedures is significant. It would be beneficial if the authors could provide a more detailed discussion on this matter.

I am happy to discuss these concerns with the authors and update my score if the questions are adequately addressed.

===post-rebuttal===

I appreciate the authors' efforts in addressing my questions and am satisfied with their feedback. I encourage the authors to incorporate the discussion from the rebuttal period into the main paper. I have raised my score to 7.

**Questions:**

Could you provide more theoretical insight on the feature extractor? For instance, is there any guidance on selecting the SSL method or determining the step size for performing the gradient step?

- Is there any method to reduce the requirement of storing the train set $D_0$?

- Could you provide more justification for Principle 1? (Please refer to the three points of concern outlined above for more details.)

**Limitations:**

I did not identify the negative societal impact of this work.

---

> ### Author Response · Authors · 2024-08-13
> **Rebuttal by Authors**
>
> We would like to thank the reviewer for these meaningful questions! Here are our responses.
>
> **Theoretical insights of SSL and the choice of gradient step size**. We have to admit that the theoretical study for feature learning is generally hard; Instead, the analysis for the SSL in the literature is through exhaustive empirical validation. One hypothesis of the SSL methods in our paper has been validated in the literature: *they can generally improve the feature representation rather than only work for specific tasks / distributions*. Equivalently, *for most data distributions, the accuracy of the classifier based on the “improved” feature representations could be higher*. Moreover, this hypothesis supports our Equation 7, which shows the loss comparison between the classifiers where *given* the test distributions the classifiers are learned with the updated feature extractor or not; we also include the empirical validation for Equation 7 in the experiment section. As analyzed in Section 3.3, Equation 7 is a sufficient condition for FLHFTL-OFU having a better loss upper bound than FLHFTL. These arguments show how the hypothesis of the SSL methods, that has been validated in the literature, can indicate our OLS-OFU to be a better algorithm than OLS in the specific task of online label shift.
>
> As for the choice gradient step size, we would have two empirical suggestions for practice. The first is to check the proper batch size for the SSL offline, for example large batch is necessary for the contrastive learning such as MoCo. The second is that as suggested in our experiment, $\tau=100$ is generally good for all three SSL methods, different datasets and different types of shifts. We recommend $\tau=100$ as a good starting point for choosing this parameter.
>
> We will explicitly add the discussion of the SSL hypothesis and the gradient step size in our revision for more intuitions behind our algorithm.
>
> **The requirement to the training set**. Thank the reviewer for raising this insightful question! We would like to first clarify that *the step of retraining the last linear layer is the only place that needs training data in our algorithm*. Moreover, as discussed in Principle 2 and line 248-252, retraining the last linear layer is designed only for three previous OLS methods (ROGD, FTH, FLHFTL); our algorithm for two other OLS methods (UOGD and ATLAS) in the literature are independent of this step. Therefore, including feature extractor updates by our algorithm for UOGD and ATLAS actually doesn’t require the training data.
>
> As for our algorithm for ROGD, FTH, or FLHFTL, we further study how the amount of training data stored for the online test adaptation influences the effectiveness of our OLS-OFU. We evaluate OLS-OFU with $0\%-100\%$ stored training data; *$0\%$ means that we still update the feature extractor but reuse the pretrained linear classifier*. The results are reported in the following table.
>
> | % stored training data | FTH-OFU | ROGD-OFU | FLHFTL-OFU | FTFWH-OFU |
> |------------------------|---------|----------|------------|-----------|
> | 100% (original)        | 8%      | 10.8%    | 7.45%      | 7.33%     |
> | 80%                    | 8.18%   | 10.93%   | 7.62%      | 7.48%     |
> | 60%                    | 8.68%   | 11.84%   | 8.04%      | 7.92%	     |
> | 40%                    | 9.49%   | 12.50%   | 8.91%      | 8.40%     |
> | 20%                    | 9.54%   | 12.63%   | 9.51%      | 8.91%     |
> | 10%                    | 9.67%   | 12.82%   | 10.11%     | 9.86%     |
> | 5%                     | 9.81%   | 12.94%   | 10.34%     | 10.03%    |
> | 0%                     | 10.24%  | 13.50%   | 10.43%     | 10.41%    |
> | OLS only               | 12.04%  | 13.65%   | 12.02%     | 11.9%     |
>
> We can observe that with less stored training data for retraining the last linear layer, the error of OLS-OFU would increase gradually. However, an important finding is that even with 0\% stored training data, which means that we keep reusing the pretrained linear classifier together with updated feature extractor, the error of OLS-OFU is still lower than the OLS without feature extractor updates. This can actually explained by the original test-time training papers [1,2], where they only update the feature extractor without refining the last linear layer and the only feature updates still brings substantial benefit.
>
> From the results, we can conclude that even if we remove the requirement of storing training data, our algorithm OLS-OFU can still outperform the OFU; more stored training data can further boost the performance. We will add this results in our revision!
>
> [1] Y. Sun, X. Wang, Z. Liu, J. Miller, A. Efros, and M. Hardt. Test-time training with self-supervision for generalization under distribution shifts. In International conference on machine learning, pages 9229–9248. PMLR, 2020.
>
> [2] Wang, Dequan, et al. "Tent: Fully test-time adaptation by entropy minimization." arXiv preprint arXiv:2006.10726 (2020).

---

> ### Author Response · Authors · 2024-08-13
> **Rebuttal by Authors**
>
> **The influence of principle 1 in practice.** We agree that empirical evidence can be important as well to illustrate the necessity for Principle 1. Therefore, we further make this ablation study to justify it: we compare OLS-OFU with its other variant named OLS-OFU-difforder where we update SSL first and run OLS later (which violates Principle 1). We compare these two algorithms across all previous 6 OLS methods and two choices of batch accumulation $\tau=1$ and $\tau=100$. The dataset is CIFAR-10, the SSL is rotation degree prediction and the shift pattern is sinusoidal shift; we observed similar performance for other settings of dataset, SSL and shift patterns and will include the full results in the reivision. The results are reported in the following table
>
> |                               | FTFWH  | FTH    | ROGD   | ATLAS  | UOGD   | FLHFTL |
> |-------------------------------|--------|--------|--------|--------|--------|--------|
> | OLS-OFU($\tau=1$)             | 11.3%  | 11.2%  | 13.9%  | 11.6%  | 11.4%  | 11.2%  |
> | OLS-OFU-difforder($\tau=1$)   | 12.33% | 12.12% | 14.35% | 12.10% | 11.91% | 12.08% |
> | OLS-OFU($\tau=100$)           | 7.33%  | 8%     | 10.8%  | 10.1%  | 8.35%  | 7.45%  |
> | OLS-OFU-difforder($\tau=100$) | 7.37%  | 8.05%  | 10.82% | 10.11% | 8.36%  | 7.48%  |
>
> We can observe that when $\tau=1$, the difference between OLS-OFU and OLS-OFU-difforder can be $0.9\%$ for OLS=FTL for example, which cannot be neglected. This means that Principle 1 indeed is important in practice when the batch is small.
>
> As for the $\tau=100$, at first it seems there is no difference between OLS-OFU and OLS-OFU-difforder. This is actually because now we only update the feature extractor every $\tau=100$ time steps; this means that Principle 1 introduces no difference between OLS-OFU and OLS-OFU-difforder for the remaining 99 steps every 100 steps. We take a closer look at the average error of OLS-OFU and OLS-OFU-difforder for only the steps where we update the feature extractor and Principle 1 has been applied. Here are the numbers.
>
> |                               | FTFWH | FTH   | ROGD   | ATLAS  | UOGD  | FLHFTL |
> |-------------------------------|-------|-------|--------|--------|-------|--------|
> | OLS-OFU($\tau=100$)           | 7.23% | 7.91% | 10.81% | 10.12% | 8.33% | 7.53%  |
> | OLS-OFU-difforder($\tau=100$) | 8.02%	 | 8.63% | 11.19% | 10.55% | 8.80% | 8.29%  |
>
> We can observe that OLS-OFU where we apply Principle 1 has non-neglecte improvements, which conclude that Principle 1 is important in practice even when the batch is large. To understand this, it is because when the feature extractor has dependency for the to-be-adapted test data, the later estimation for the distribution of these test data in the OLS method can be more inaccurate and hence this can hurt the performance.
>
> Overall, Principle 1 is important not only just for the theoretical results, but also matters in practice. We will add this further analysis to our experiment section!

---

> ### Author Response · Authors · 2024-08-13
> **Rebuttal by Authors**
>
> We hope we were successful in addressing your concerns. Please let us know if you have any additional concerns. We look forward to hearing from you!

---

### Decision · Program_Chairs · 2024-09-25

**Decision:**

Accept (poster)

**Comment:**

This paper studies the problem of online generalized label shift adaptation, proposing a feature learning stage integrated with existing methods for online label shift. The authors theoretically demonstrate the soundness of their methods and conduct a variety of experiments to show their practical effectiveness.

The paper initially received mixed evaluations, so I invited another expert reviewer. After checking all the reviews and also paper myself, I find the paper's contributions interesting and valuable to the community. Specifically, integrating the feature learning (i.e., the representation ability of deep neural networks) into online label shift adaptation is highly useful for practical applications. Therefore, I recommend the acceptance. However, several reviewers have raised constructive feedback, including suggestions to enhance the experiments to make their claims more convincing. Moreover, it is also important to discuss the storage complexity in the initial stage. Other interesting issues include how to manage new classes during the online adaptation phase, etc. The authors are encouraged to take those comments into account when revising the paper. Some issues may be acknowledged as limitations or noted for future work. It is common in scientific research to focus on certain points and leave some limitations in future studies, but it is important to discuss and acknowledge them clearly.